# How to adapt caring services to migration-driven diversity? A qualitative study exploring challenges and possible adjustments in the care of people living with dementia

**Mette Sagbakken**[1]*, **Reidun Ingebretsen**[2], **Ragnhild Storstein Spilker**[3]

**1** Department of Nursing and Health Promotion, Faculty of Health Sciences, OsloMet - Oslo Metropolitan University, Oslo, Norway, **2** Norwegian Social Research (NOVA), OsloMet - Oslo Metropolitan University, Oslo, Norway, **3** Unit for Migration and Health, Norwegian Institute of Public Health, Oslo, Norway

* metsa@oslomet.no

**Data Availability Statement:** The datasets (transcripts) generated and analyzed during the current study are not publicly available due to risk

## Abstract

### Background

Research on how services can be adapted to meet the needs of people with dementia with an immigrant or minority ethnic background is scarce. Several approaches have been discussed: offering services adapted to language and culture, adding bilingual staff to mainstream services, and providing cultural awareness and sensitivity training to health personnel in mainstream services. This study seeks to develop more knowledge of challenges and possible adjustments related to receive and provide public care for people living with dementia with an immigrant or minority ethnic background.

### Methods and materials

Through a qualitative design, including 19 single interviews, 3 dyad interviews and 16 focus groups with older immigrants, relatives of immigrants with dementia, and health personnel, we explored experiences and perceptions related to receive and provide care for people with immigrant backgrounds living with dementia in Norway. The analysis were conducted inspired by Kvale and Brinkmann's three contexts of interpretations.

### Results

Challenges related to language and communication were emphasized as the most fundamental barrier to provide adjusted care; exemplified through cases of isolation and agitation among patients not able to communicate. Care services framed by the majority culture creates feelings of alienation and exclusion. Not having access to specific types of food and the possibility to listen to songs, music, literature or TV programs representing a familiar and homely context may prevent use of public dementia care. Findings also point to differences in moral views regarding life-prolonging treatment in advanced stages of dementia.

of recognizing the participants. Additional quotes and examples, that will support the findings, can be provided upon request to Unit for Migration and Health, Norwegian Institute of Public Health, P.O. Box 222 Skøyen, 0213 Oslo | Phone: + 47 907 64 709 | www.fhi.no.

**Funding:** The study was supported by the Norwegian Directorate of Health. The financial sponsor played no role in the design, execution, analysis and interpretation of data.

**Competing interests:** The authors declare that they have no competing interests.

## Conclusion

This study argues that to be able to address challenges related to migration-driven diversity one needs holistic care services that addresses individual as well as socio-cultural needs. A linguistically and culturally diverse workforce may represent an important resource, potentially reducing some of the problems related to communication. On a structural level, it seems necessary to allocate more time and resources, including the use of interpreters, when assessing and getting to know persons with dementia with another linguistic and cultural background. However, shared language does not guarantee understanding. Rather, one needs to become familiar with each person's way of being ill, on a cultural and individual level, including changes occurring living with progressive dementia. Getting to know a person and his/her family will also facilitate the possibility to ensure a more familiar and homely context. Thus, continuity in relation to language and culture is important, but continuity in relations may be equally important ensuring that people with dementia receive equitable care.

## Introduction

### Ageing in unfamiliar landscapes

Norway is becoming increasingly more ethnically, culturally and linguistically diverse. Immigrants account for 14.7 percent of the country's population, and those who are Norwegian-born to immigrant parents' account for 3.5 percent of the population. Residents of today's Norway come from 221 different countries and independent regions [1]. The immigrant population is, on average, considerably younger than the general population: only 4 percent of persons in Norway older than 70 years are immigrants. However, according to Statistics Norway, this number is projected to reach 27 percent in 2060 [2]. This poses challenges with regards to health and care services on how to adapt and organise their services in order to serve a more diverse group of older people.

The Global Alzheimer's and Dementia Action Alliance [3] emphasizes that dementia is a global public health challenge. Indeed, dementia is described as the greatest global challenge for health and social care in the 21st century [4]. In Norway, a country with 5.3 million inhabitants, it is estimated that approximately 80,000 people are living with dementia and that approximately 10,000 persons develop the disorder each year [5]. With the ageing of the population, the number of people affected by dementia could double by 2040–2050 [6]. Age related cognitive impairment (not diagnosed) or dementia causes extensive use of health and care services in Norway. More than 80 percent of those living in nursing homes and more than 40 percent of people > 70 years receiving home-based services have some sort of cognitive impairment or dementia [7]. In recent years, there has been an increasing demand for personnel in health and care services in Norway. This demand is partly due to the increase in the number of older people living with complex clinical conditions, including the increasing amount of people with dementia. International recruitment of nurses and medical doctors has been one way to meet this challenge. Another solution has been to recruit and train immigrants living in Norway to work in nursing homes and home care services [8].

Immigrants' experiences of aging and health are diverse as they belong to different socio-economic, ethnic, cultural and linguistic groups and have varying levels of education and work experience. Further, they have various reasons for migrating, and have lived in Norway for

different lengths of time [9]. Family structures and relationships are diverse both within and between different immigrant groups. However, many older immigrants come from societies where family structures and expectations of care from family members are stronger than in Western societies, often involving strong normative feelings of attachment, responsibility, and reciprocity [10, 11]. For immigrants from low- and middle-income countries, there has been a shift from high mortality and high birth rates to low mortality and low birth rates. This has increased the actual number of living generations and decreased the number of relatives that can live together with their extended families [12]. Additionally, immigrant families may be separated or divided during the migration process, which can make 'traditional extended family care' difficult to provide. Last, the competing demands associated with different family members taking on new roles in new settings can make it challenging to allocate time to care for family members who become severely ill [12]. Thus, even though few older members of immigrant groups live in care facilities in the Nordic countries [13], this pattern may change as an increasing number of immigrants become older and potentially develop dementia. However, current care services are undoubtable designed for a more homogenous group of people, and it is argued that this poses challenges to equitable services [14].

Ethnic differences in the use of dementia care services have been documented internationally [15–17], as well as in Norway [18]. Different studies show that various minority ethnic and immigrant groups' access diagnostic services at a later stage, are prescribed and use anti-dementia medication to a lesser degree and are less likely to receive care in a nursing home [15–18]. There are several potential explanations for these differences, such as cultural differences regarding caregiving and family networks, perception and recognition of dementia and associated symptoms, negative experiences with health and care services, and language barriers. Another reason for the underutilization of services may be that the service needs of minority ethnic and immigrant groups are unidentified and therefore unmet; thus, the services are not as accessible to all patients groups [14–16]. Several studies show that many older people with minority ethnic and immigrant backgrounds and dementias are living within extended family networks and tend to use the formal services of dementia care to a lesser extent than the majority population [17, 19]. Patterns of family caregiving are found to be shaped by factors such as lack of awareness of public services; barriers related to availability and accessibility; stigma associated with dementia and stigma related to the use of formal services; language barriers; and viewing existing services as culturally inappropriate [19, 20]. Seeing services as culturally inappropriate relates not only to issues such as food habits or different customs but also to concepts such as filial piety [19]. This concept refers to a strong identification and solidarity with family members and involve strong normative feelings of responsibility and loyalty in taking care of and being good to one's parents [21]. For example, a study on dementia caregiving among Iranian immigrants in Sweden found as a prevalent perception that dementia could be caused by a lack of social relations, social interaction, and care; thus, the shame associated with not fulfilling the standards of filial piety was strong. This perception made it difficult to receive help from any type of formal services [22]; findings similar to studies among minority ethnic groups in a variety of contexts [17, 20, 23, 24]. Thus, perceptions about causality combined with strong cultural norms of family care may cause people with minority ethnic and immigrant backgrounds to use relevant services either late or not at all [20].

There is little research on how services could be developed and adapted to meet the needs of people with different linguistic and cultural backgrounds living with dementia. Several approaches have been discussed in the literature: offering special services adapted to language and culture, adding bilingual staff to mainstream services, and increasing the cultural awareness and sensitivity training of health personnel in mainstream services [25]. Adding bilingual staff seem to be particularly important as persons with minority ethnic background often face

stronger problems with communication due to a lack of knowledge of, or dementia-related loss of a second language [26, 27]. Another dimension of this, is that language and culture are suggested to strongly overlap, recognised among others through the principle of *linguistic relativity*; a concept that illustrates how the structure of the language is shaped by the speakers' world view or cognition [28]. Thus, the way of speaking, such as the use of certain words, linguistic categories, or by the use of implicit references to religious and culturally shared knowledge (for example well known historical events, fairy-tales, and songs) will influence the ability to understand one another [26].

**Aim of the study.**   This study seeks to provide more knowledge on challenges and possible adjustments related to provide public care for people with immigrant backgrounds living with dementia in Norway. This topic is approached from the perspectives of health personnel, relatives of people with dementia as well as older people with an immigrant background and seeks to explore dimensions related to both culture and language in different care settings.

**Use of concepts.**   In this paper, we use "minority ethnic groups" when we refer to studies from outside Scandinavia, as this is the most commonly used term in these papers. However, in the parts of the manuscript where we refer to Norwegian statistics, Norwegian/ Scandinavian studies, and the present project, we use the term "Immigrant", which is the terminology used by Statistics Norway, and is the most commonly used term in academic and public discourse in Norway. Importantly, the use of "Immigrant" also indicates that we are not referring to our native population, such as the Saami's.

## Materials and methods

This study is part of a comprehensive study on older immigrants and dementia in Norway, where the overall goal has been to assist the Norwegian Directorate of Health in designing appropriate strategies for the care of immigrants with dementia. Target groups for this project are immigrants above the age of 50 years, relatives of immigrants with dementia (family caregivers), health personnel and care workers, and decision and policy makers.

Since research on the care needs of people with dementia with an immigrant background in Norway is limited, the research team considered qualitative methods to be the most appropriate approach. The chosen theoretical underpinning was phenomenology; a methodology that attempts to understand the meaning of events and interactions within the framework of how individuals make sense of their world. Through qualitative individual and dyad interviews as well as focus groups discussions, we sought to gather the different participants' expectations, views and experiences regarding receiving and providing care to people with dementia who have an immigrant background. We did not seek the views of groups from particular ethnic (or religious) backgrounds. Rather, the position of interest is that of being a person with a different linguistic and cultural background in need of care due to cognitive impairment (not diagnosed) and dementia; and the experiences of those providing relevant services.

The research project has already published 3 papers; one paper explored the perceptions of dementia and cognitive impairment among healthy older immigrants and relatives of immigrants with dementia [29]; another paper explored the experiences of health personnel with regard to identifying, assessing and diagnosing immigrants with dementia [30], while the third paper focused on different care patterns among immigrant groups [31].

### Sample strategy

To be able to gather views and experiences from a variety of relevant respondents, we applied a purposeful sampling strategy aimed at variation [32] in regards to factors such as gender and ethnic background among older persons and relatives of persons with dementia, and

professional background, workplace, type of experience, and ethnic background among health personnel being interviewed.

**Older immigrants.**   The overall study was initiated by conducting nine Focus Group Discussions (FGDs) with 51 healthy older immigrants from 10 different countries; exploring views and experiences regarding accepted and common treatment and care practices to people with age-related cognitive impairment or dementia. We obtained a purposeful sampling strategy, aiming at including many different perspectives and experiences. The participants were recruited, by the second author, from a variety of interest organizations for immigrants, religious communities and senior centres. The final sample consisted of 34 women and 17 men between 50 and 80 years of age originally from Pakistan, India, Afghanistan, Iran, Turkey, Algeria, Mexico, Chile, Poland, and Bosnia.

**Relatives to family member affected by dementia.**   In Norway, only 4 percent of those above 70 years are immigrants [2], and few older members of immigrant groups live in care facilities in Norway [13]. Thus, it was difficult to identify and recruit relatives to persons living with dementia, and we had to rely on convenience sampling. With the assistance of health personnel in diagnostic clinics at hospitals, nursing homes and day care centres or community-based home care, 12 relatives between 25 and 78 years old who had family members affected by dementia were recruited for in-depth interviews (IDIs). Among the relatives, there were 10 women and 2 men; six of the women and both men were children of the person living with dementia (aged 25–55), and the remaining four women were spouses (aged 65–78). The relatives originally came from Afghanistan, Pakistan, China, Vietnam, Turkey, Lebanon, Sri Lanka, and Chile.

**Health personnel.**   To gain knowledge of the views and experiences of health personnel working with people from immigrant groups, 18 health personnel (mainly nurses and nursing assistants) were recruited from community-based home care services and nursing homes in Oslo. During this process we found it difficult to find health personnel who had direct experience with patients living with dementia representing another linguistic and cultural background (see reference [2, 13]). Thus, also in this case we had to rely on a convenience sampling; using already existing network within local care services that we knew had patients with immigrant background and dementia. These service providers, 15 women and 3 men, in the age range of 35 to 55, participated in four FGDs. Later in the research process, in order to reach saturation, we broadened our sample and made a purposive selection of 27 health professionals; 18 women and 9 men in the age range of 38 to 62, originating from seven different countries (Sweden, Ethiopia, Sri Lanka, Morocco, Philippines, Pakistan and India) in addition to Norway. To be able to access participants who had experiences with older immigrants with age-related cognitive impairment or dementia, we combined an approach where we sent an information letter about the study to all GP centres in four districts of Oslo with a high proportion of immigrant populations, and through using existing networks and contacts. When recruiting in other parts of Norway, we sent information letters and called persons and institutions that seemed relevant; such as nursing homes and home-based services in areas with significant immigrant populations. The participants represented different parts of Oslo as well as six different counties in the northern, western, and southeast parts of Norway, and they participated in three FGDs, seven IDIs, and three dyad interviews (DYI). The participants were GPs, medical doctors, nurses, auxiliary nurses, and leaders of nursing homes, and they worked in GP centres, nursing homes, short-term nursing homes, day care centres, home-based services, geriatric polyclinics, psychiatric polyclinics, "memory clinics" (diagnostic departments in hospitals), a community health centre with services for refugees, or were part of a primary health care dementia team.

## The research context

Norway has a tax-financed public health-care system, and the maximum fee for health services is 230 euros per year (excluding dental care). Public sources account for more than 85% of the total health expenditure, and most private health financing comes from household out-of-pocket payments. Thus, in principle all persons registered in Norway (including all immigrants who are legal residents) have equal access to health care such as services provided by the general practitioner (GP), hospital services, homebased services and nursing homes. The municipalities are responsible for primary health care and social services, including the provision of care and practical help for older persons receiving day care or care at home, or for those living in nursing homes [33].

## The research team

The research team consisted of three female researchers with different backgrounds. The first author (MS) is a registered nurse and a professor in global, public health. The second author (RI) is a researcher and a specialist in clinical geropsychology, while the last author (RS) is a registered nurse with an MPhil in Health economics, policy and management. The authors were responsible for conducting all the interviews and focus group discussions.

## Data production

The initial FGDs with healthy older immigrants lasted 90 to 120 minutes and took place at their respective organizations or at the researchers' workplace. In one of the FGDs, a research assistant speaking the participants' language moderated the group, and the researcher was present. In five of the FGDs, the group discussions were conducted in Norwegian because the participants spoke adequate Norwegian. An assistant who spoke the mother language of the participants was, however, present to help translate and explain when needed. In the remaining four focus groups, all spoke good or fluent Norwegian. All the FGDs consisted of five to eight participants and the interviewer encouraged a discussion around each of the introduced themes. Questions related to perceptions regarding aetiology and common and accepted treatment practices for people living with dementia, were addressed using a semi-structured interview guide. The questions in the interview guide were all inspired by a thorough literature review that was part of the overall study commissioned by the Norwegian Directorate of Health. The guide contained relatively few themes to allow time for in-depth discussions and for new themes to be discovered.

The semi-structured IDIs with family caregivers lasted 60 to 150 minutes and were held in their homes, at the research centre (NAKMI), or at the nursing home where their relative resided. Family members who did not speak Norwegian were offered an interpreter. Questions related to how the family had managed the symptoms and disease from initial symptoms until the present situation and how and to what extent they had initiated help and cooperated with existing services. Their views on best possible care and responsibility for care, including potential specific concerns related to family members with dementia and a different cultural and linguistic background, were addressed.

The IDIs, DYIs, and the FGDs with health personnel (five to eight participants) lasted 60 to 120 minutes, were conducted in Norwegian and took place at the participants' workplace. A semi-structured guide was used, and the questions were to a certain extent adjusted to the profession and position held by the participants. The aim was to gather descriptions regarding participants' perspectives and experiences in relation to the assessment, treatment and care needs of persons with dementia representing immigrant groups and to interpret the meaning of these descriptions.

The questions in these interview guides were inspired partly by a literature review, partly by perspectives provided by the first FGDs with older adults with immigrant backgrounds, and partly by discussions of experiences with experts in the field (health personnel, researchers, nongovernmental organization representatives) who served as a resource group throughout the research period. The approach was flexible in the sense that the interviews and discussions were guided by answers and themes introduced by the participants. All the data were tape-recorded and transcribed verbatim.

## Analysis and interpretation of text

The analysis was not an isolated process, but rather a continuum starting from the beginning of the study until its completion. To optimize the analytical process, all three researchers wrote descriptive and reflective logs. These logs, among others, identified emerging themes and informed and inspired the ongoing analytical reflections and discussions. Data from the initial FGDs with older immigrants from 10 different countries were analysed and discussed before the interviews were conducted with relatives and health personnel; thus, these data served as inspiration for identifying themes and questions in consecutive interviews (IDIs and FGDs), and helped contextualize data in the following interviews. All the material was analysed and interpreted following the general principles of Kvale and Brinkman's [34] descriptions of levels of interpretation: 1) Self-understanding, 2) Critical common sense understanding, and 3) Theoretical/abstracted understanding. These three interpretational contexts derive from different explications of the researcher's perspective, lead to different levels of analysis and serve to make explicit the analytic questions posed to a statement. Thus, aiming at reproducing the participants' own understanding (level 1), the interviews and focus groups were transcribed verbatim after the interviews were conducted. Then, two of the co-authors read the transcribed interviews in their entirety to acquire an overall impression of the content. This process involved searching the entire body of material for patterns and deviances and for similar and contrasting statements. Units of meaning, inspired by the study's aims and by discussions between the authors, were identified by color-coding. Based on discussions related to which themes each units of meaning represented, the researchers formulated what the subjects themselves understood to be the meaning of their statements in a condensed form. The next step (level 2), which involved a critical common-sense understanding, included a condensation of the wider frame of interpretation. By adding a general understanding about the content of the statement we widened the scope, thus amplifying and enriching the interpretation of the participants' statements. This part of the analysis generated preliminary themes by labelling the paragraphs and sentences with sub-themes. In the last phase (level 3), the sub-themes were linked together and described through central themes reflecting the objectives of the study. This more comprehensive interpretation involved contextualizing the critical common-sense understanding using previous research and to some extent theory, thus moving our analysis to a higher level of abstraction (34), as illustrated in Table 1 below:

## Ethics

The project was registered and treated by the data protection officer (personvernombudet) at Oslo University Hospital and the Norwegian Social Science Data Services (NSD), and it was granted permission before the data collection started. The project was submitted to the Regional Committees for Medical and Health Research Ethics (REC) South East but was considered to be outside the remit of the Act on Medical and Health Research and could therefore be implemented without the approval of the REC. The participants were informed about the

**Table 1. Example of the analytical process.**

| 1. Self-understanding | *"It is obvious that those immigrants that are here (nursing home) are the most isolated. . . they have no one to talk to often, and just sit there without being able to communicate. . ."* |
|---|---|
| | *"It often happens, when there are frustrations, when they are not able to understand the language (anymore) [. . .], or when they are not able to speak Norwegian, then it develops into aggression . . . acting-out"* |
| 2. Critical common-sense understanding | Not being able to communicate causing isolation and frustration: Respondents told how not being able to talk in the majority language or another shared language seemed to represent a source of isolation, loneliness, aggression/ agitation among patients; thus also a challenge for health personnel in providing good care. Several mentioned that some of the patients with immigrant background did not have relatives/social network; thus being particularly vulnerable in relation to becoming isolated. |
| 3. Abstracted/theoretical understanding | Studies on communication across linguistic and cultural diversity show that residents who do not speak the majority language tend to have less communication with the staff compared to people representing the majority group (38, 39) and tend to sit more alone (37). The relationship between agitation and people with dementia who do not speak the majority language is also thematised (43); patients having the majority language as the second language are associated with significantly higher agitation level. Such a position can be seen in the light of intersectionality theory (54) as lack of exposure to the majority language and culture; including a lack of social network are likely to be interlinked with migration history; ethnicity; level of education; job opportunities/experience; gender roles; thus creating differences in the possibility of accessing adjusted care. |

purpose of the study and that they could withdraw their participation without giving any reason. All participants gave written consent to participate in the study.

## Results

### Receiving care in unfamiliar surroundings

One of the topics that was frequently raised by health personnel as well as relatives was that many of the patients who were placed in nursing homes or used day care centres were not familiar with these types of care institutions. Additionally, many of the immigrants living with dementia and their relatives had maintained their cultural traditions and practices, and moving into- or spending the day in an institution framed by the majority culture could create a feeling of alienation. One of the relatives pointed to this by underlining how *"sharing food, tastes, smells, sounds, language and music contributes to build communities"* and how the presence or absence of such references would be interpreted as *"whether one belong or not"*. Another relative described how she, despite needing relief, declined the offer of day care for her mother at a centre for people with dementia. Exploring the facilities, she found that her mother would be the only one with an immigrant background, and she concluded that her mother would not feel at home there. Yet another daughter told that she took her mother out of the day care centre because her mother was the only one with immigrant background and she felt that she was not included in the community:

"*This generation of the (Norwegian) older people . . . it is probably going to change. . .they are not very inclusive, I felt it went very wrong.*"

Also among the participants in the FGD's with older immigrants there were examples of relatives that had avoided asking for formal care because they were afraid that the service given

was not linguistically or otherwise adapted to their needs. Similarly, also in this group there were examples of relatives that had stopped using public services because they felt that the help the person with dementia received was not "fitted" to their needs; either due to language barriers, or a feeling that their family member became isolated in the prevailing majority culture of the institution.

Relatives also offered several examples of how they had tried different types of public help but that they ended up "being on call" for different reasons, such as having to assist due to language barriers or due to the older person refusing to accept the food or the care provided by others. A daughter tells about her experiences with her father:

> "*We have tried a day care centre and short-term stay at a nursing home, but we could not leave him there. When he was there (nursing home), we shared the time in between us (sisters), one came in the morning and the other came later to relieve. . . [. . .], so we more or less did everything. After two days, we tried to avoid being there the whole day, but then they (the staff) did not manage to feed and give him the personal care he needed . . .*"

In this example, the daughter found the staff to be kind and to be doing their best, but as they were unknown to her father and unable to speak his language, he was afraid and unable to relax. They decided to take him home.

Health personnel shared similar experiences, relating both to relatives who found it difficult to let go of responsibility and to patients who could not thrive due to other habits and customs. A GP with an immigrant background herself elaborates:

> "*. . . and then there is this question about integration. How are we to integrate these immigrants [in nursing homes] who are used to only being at home . . . and where they always have their family around them. . .? Suddenly they are to meet other people. That is not that easy. . .*"

As stated in the quote, being accustomed to mainly relating to family members, particularly when those relations involve cultural or religiously based rationalities of thought and behaviour, care preferences and food habits, would make it difficult to provide adjusted care within the frameworks of the nursing homes' or even day care centres.

## Receiving care within the frames of unfamiliar care practises

Another finding of interest, often addressed by health personnel, was that some of the relatives had a different view on how care was to be provided in final stages of life. In all the cases mentioned, the unifying idea was that it was up to God (Allah) to decide when a person was to die, and up to that point one was to use all available means to preserve life. A nurse in a FGD elaborates on one of her experiences related to food intake:

> "*They (relatives) said they wanted to keep their father alive, no matter what it took. It was mainly related to the issue of nutrition . . . that he had to have this and that. . .*"

One of the relatives being interviewed related his involvement in the care to how his religious and ethical views on life-prolonging treatment were in contrast to the view among the health personnel in the nursing home, who encouraged him to allow his father to die "in peace and quiet". Although his father was very ill, being in a late stage of dementia, he still wanted to secure him the best possible medical help:

"*He is my father. I know he had a strong will to live. . . Now it is my responsibility to secure him the best possible help. . .*"

There were several instances where different ideas around life-prolonging treatment led to conflicts between health personnel and relatives (and sometimes between relatives), and where the county administrator (Fylkesmann) and/or the relatives' trusted Iman were involved in trying to resolve disagreements. The discussion mainly focused on whether one could use force to secure nutrition in late and terminal stages of care. A nurse elaborates on a case where the patient, living with dementia, was in a terminal phase:

"*She (relative) sat there the whole night and held his hands while he was sleeping. She was awake . . . to make sure that he did not pull out the tube nutrition. . .*"

Other health personnel mentioned that such situations or similar types of situations could also conflict with another value-loaded idea that they often encountered: avoiding the use of painkillers and sedatives as part of (terminal) care–even though the patient (in their view) needed such medication to be able to sleep, to calm down, and to reduce pain.

In general, relatives' strong involvement in this type of treatment and care decision-making was described by health personnel as difficult to relate to, as it involved ethical as well as juridical dilemmas.

## The importance of familiar food

Health personnel, as well as individuals in the FGDs with older immigrants, frequently mentioned food habits as one important issue, even though specific food requests, such as access to *halal* food, tended to be fulfilled. Participants in a FGD in a nursing home, who were experienced with having residents with an immigrant background, used time to discuss the topic of food:

(OBJ 1) "*. . .and then it is the food culture, which is very important. . . So many are used to eating really spicy food, and then they get (Norwegian) fish balls in sauce. . . [. . .]*"

(OBJ 2) "*They do get halal food, but the food is not varied [. . .] It is the same type of food that is offered all the time, like halal sausages or. . . [. . .] It has no taste in it, it is nothing compared to what they are used to. . .*"

Thus, even though efforts are often made to adhere to important food habits, there was a general concern about food, which led some relatives to bring their own food. Health personnel described relatives as a resource in regard to food, as many patients would not eat or would forget to eat if they were served something they did not like. However, when relatives brought food, this would sometimes conflict with the routines and the rules of the nursing homes, as it was difficult to get an overview of the food intake of the patient because the pattern of bringing food was often inconsistent. Some of the personnel also claimed that bringing food from the outside was not in line with the Norwegian Food Safety Authority. Several participants, across the groups of respondents, raised the issue of adjusting the nursing homes in a way that allowed relatives to prepare food for their family members. One of the doctors at one of the nursing homes suggested the possibility of allowing relatives to cook at the nursing homes:

"*It would be nice if relatives were able to cook some more food themselves. I would like that. A new nursing home should take it into account. . .*"

In homebased services the difficulties in providing adjusted food was also often raised. Due to limited resources, the nurses could only offer food being easy and fast to make; often implying typically Norwegian "fast food" such as eggs or slices of bread. A nurse refers to a typical example where they had offered to help with preparation of food to relieve relatives to a person with dementia:

"*We talked about providing food such as "Fjordland" (ready-made dishes to be heated), sandwiches or eggs, and that no, we could not prepare anything else . . . and then they withdrew. Because it does not correspond with their culture and then they rather provide the services themselves. . ."*

## Providing and receiving care in unfamiliar languages

Even though some topics were referred to as cultural differences that were sometimes difficult to relate to, most of the respondents (health personnel, relatives and older immigrants) tended to describe language and thus communication as the main challenge in providing adjusted care for people with dementia with immigrant backgrounds.

According to different participants, many of the patients with dementia representing immigrant groups do not speak Norwegian or have lost or partly lost the Norwegian language they have acquired as their second or third language. Some used a combination of languages that only the close relatives could understand, making the staff dependent on the relatives to understand the patient. Lack of a common language between the person with dementia and the formal carers made it challenging to adjust care; described by the health personnel as difficulties in identifying and assessing the (often changing) care needs of patients. Further, language difficulties made it difficult to understand whether there was a general misunderstanding or whether there was a development in the clinical picture. One of the nurses exemplified how changes in the ability to communicate normally would alert health personnel to consider clinical changes. However, when the language did not provide such information, such changes could be overlooked. A nurse elaborates:

"*If we say something to a Norwegian and he does not understand, we will wonder more about what is wrong. In the case of immigrants, we may think that it is "the language", and that we said something he did not understand, and we probably do not investigate further. . ."*

Due to budget constraints, interpreters were seldom used, even during the admission process to a nursing home. The same was described in homebased services; health personnel reporting that they were not assigned more time nor resources (such as interpreters) during assessments or the regular home visits. Many emphasized the difficulties in finding ways to get to know the person living with dementia, and that they had to rely more than usual on relatives (if accessible). An experienced nurse assistant in home-based services elaborates the challenges related to care for persons who did not speak Norwegian and whom they did not really know:

"*It is difficult sometimes, if they are alone when we arrive (at home). . . What do they want to eat, we don't know that. . . No, they don't like bread (Norwegian sandwich), they don't like this and that, and then it becomes like [. . .] but what do you want then? [. . .] Then, it develops into . . .'leave me alone!'"*

(*Patients expressing that they want the nurse assistant to go*).

Others mentioned patients they knew who had been at the nursing homes for years without being able to communicate with health personnel or other residents. This was particularly relevant for some of the immigrants who had come alone to Norway and who had never been reunified with their families. A doctor at one of the nursing homes talks about these situations:

"*It is obvious that those immigrants that are here (nursing home) are the most isolated. . . they have no one to talk to often, and just sit there without being able to communicate. . .*"

Both health personnel and relatives described not being able to talk to patients in a shared language as a source of loneliness potentially causing depression, anxiety, aggression and agitation. Responding to a question about the challenges of caring for nursing patients with a different cultural and linguistic background, one of the nurses in a FGD with health personnel from a dementia ward in a nursing home captures what seemed to be a prevalent understanding:

"*It is the language that is. . . that plays more. . . (of a role) than culture in a dementia department. [. . .] It often happens, when there are frustrations, when they are not able to understand the language (anymore) [. . .], or when they are not able to speak Norwegian, then it develops into aggression . . . acting-out [. . .] Today I had one (referring to one patient with dementia) that has big problems because he only speaks. . . (European language) [. . .] He does not understand . . . he thinks everyone knows his language. . . that is what he believes. . . And he expresses that he thinks that no one cares, and that he is left here [. . .], and then he sits and cries or he lies down on the floor kicking about.*"

Health personal described different ways of trying to manage the language barrier, such as the use of "Google translate". One nurse assistant in a nursing home described how this might help in some situations:

"*We have a tablet computer in the ward, so we can use that one. We just talk on that application and it translates. [. . .] it is not totally right (the content) but it helps a little. . ., it helps. . .*"

Others described using colleagues who were able to speak the person's language and how that would help to relieve difficult situations. Some of the nurses or nurse assistants were even called on the phone when off-duty to help calm down agitated patients who spoke a language the others did not understand. A nurse assistant tells about a situation he frequently encountered:

"*There is this patient; he has lived in Norway more than 40 years, so he can speak English and Norwegian. [. . .] Now he is getting anxious, because he is losing all the languages. So. . . Urdu helps him now. [. . .] When he is agitated or frustrated he only speaks Urdu. [. . .] So I have to go to him to calm him down.*"

Some of the health personal being interviewed described how they tended to speak Norwegian to patients who had another primary language and who did not speak Norwegian. Although it did not *"feel right"*, it was somehow easy to do because the patient did not respond. Others described how they went around with different miniature dictionaries or some words on a list, which they used in situations where patients needed to be calmed down. In general, language and related communication barriers were described as the most important barrier in

preventing aggression and acting-out behaviour and to providing good treatment and care in general.

## Lack of continuity of care

As emphasized by health personnel as well as relatives, language barriers were often combined with another important and related dimension; lack of continuity of care. A patient's daughter, receiving help from home-based services, exemplified this:

> "*He got help with showering, theoretically yes, but with different people all the time. Someone spoke bad English, some bad Norwegian [. . .] There was simply no platform for cooperation . . . no continuity. Without continuity, comfort is not given; dementia makes it impossible to have new people coming all the time. They thought that my father was hard to help. . .*"

As pointed out by the relative, getting to know an individual patient and his or her ways of verbal and non-verbal expression demands continuity, not only (verbal) language knowledge. Due to language barriers and a lack of established and continuous relations with trusted health personnel, some relatives reported that they felt forced back into the role of the main caretaker.

The need for continuity was also supported by health personnel and was frequently discussed in interviews and focus groups. In one of the FGD's involving homecare nurses, continuity was linked to the possibility to get to know the person; including his/her body language:

> "*Most of the patients have relatives, and you have to talk to them too (to get to know the patient) [. . .] and then you will observe over time. . ., because if you already know the patient a bit, then you will manage to see if the situation is not ok. . .*"

When asked what type of knowledge and competencies health personnel needed, the discussion in one of the FGD's with health personnel turned into a focus of the need of continuity of care:

> "*There is a need for more continuity [. . .] so that there is not that many to relate to, because then it becomes difficult (for health personnel) to identify needs. . . (group members nodding). And it is also difficult to receive help when you don't know the one that is coming. If you are to take care of those people . . .. and, even more in relation to people from other cultures. . .. you need more continuity and more time.*"

## Views on culture-specific care

One of the questions posed to all the participating health personnel was whether they, based on their experiences, believed that there was a need for nursing homes adjusted to specific cultures. With a few exceptions, nurses, nurse assistants and doctors objected to that idea, arguing that bringing people together based on culture and language would go against the ideal of integration and inclusion and potentially feel discriminating. Several participants mentioned that one should avoid "*ghettofication*" or "*segregation*" in nursing homes and that the diversity of Norwegian society should also be represented in the nursing home context.

In the FGDs with older immigrants, the participants first of all underlined the importance of being able to speak their native language when discussing the need for adaption. Different types of language support and to have staff with linguistic and "cultural competence" in both

home services, day care centres and nursing homes were seen as important. Health personnel also addressed the need of further competencies, and related this to more knowledge on different ways of understanding and handling dementia; including different perspectives on dignified ways of dying or rationalities behind life-prolonging treatment. A manager at one of the nursing homes emphasized the need for a broad type of competence:

"*It says in the guidelines for nursing homes, that we are to adjust for people's spirituality needs and practices. That means that we have to have knowledge of other cultures and other religions. However, in Norway, there has been a tendency to focus on value neutrality. . . but we should include more openness to different philosophies of life. . . [. . .] In general, people working in the health sector need more knowledge of how to relate to opposing philosophies of life. . ..*"

According to this participant, addressing diversity in monocultural institutions would imply knowledge of–as well as acceptance and facilitation of–how to integrate other philosophies and religious practices. Thus, institutions should be *open* instead of *neutral* in their approaches to different patients.

Even though most of the participants did not seem to embrace the idea of culturally profiled services, many of the older immigrants as well as relatives expressed a need for more staff from the residents' countries of origin to be part of the services. A person in one of the FGDs with older immigrants states:

"*. . . when nurses are from the same culture . . ..it is easier to understand both culture and language.*"

As emphasized above, coming from the same country may facilitate understanding of linguistic expressions but also an understanding of that person's historical and cultural background. A few participants saw the integration of staff with varied ethnicities or language backgrounds as an entitlement, emphasizing that older patients with dementia, should be entitled to talk to someone who understood them; thus, these participants supported the idea of culturally profiled nursing homes. A doctor with an immigrant background herself explains her thoughts:

"*A Pakistani patient should be allowed to talk with, communicate with, a Pakistani doctor. A Somali patient should be able to talk with another Somali patient . . .and doctor. . . [. . .] In the last period of life, you should be able to gain that type of respect, that honour, that dignity. In a way, this is about them being allowed to keep that.. [dignity). If they forget things. . . and we think "well, they don't understand anything anyway. . .", that is wrong. . .*"

As illustrated above, those who favoured care services with a cultural profile emphasized the importance of language, i.e., the importance of being understood and thus preserving adapted and dignified care. Others mentioned the need to move away from the idea of monocultural nursing homes, in the sense that only the majority culture is represented through the food or activities being offered. Across the groups of respondents, many participants suggested that instead of separating people from different cultures, one should adjust to culturally specific needs within the frames of the care services. This could be done by providing room for different religious activities, international TV channels, wider variety and individually based food and care, larger rooms allowing many family members to visit at the same time, and by allowing international or individually based music to be played in one's private room.

Some of those who emphasized the need for nurses, doctors and nurse assistants who represented diversity in language as well as cultural backgrounds related this need not only to variety in language and cultural competencies but also to what some of the participants described as a more dedicated trust in people representing another culture. A nurse assistant in a FGD elaborates:

> *Interviewee*: "*No matter what, being a patient with dementia here (nursing home), the way I experience it, independent of being from Pakistan or Afghanistan, they see that I am from. . . (pause)*
>
> *Interviewer*: "*Another culture*?"
>
> *Interviewee*: "*Yes. They call me more often than the others (nurses). It is natural. . .*"

This participant, as well as other participants among the health personnel, pointed to a trust between patients and health personnel from the same language and region or culture. However, as indicated in the quote, the experience also seemed to include some type of trust between the patients and health personnel who did not represent the majority culture.

## Discussion

This study shows that there are needs related to language or culture that appear challenging for health care personnel to meet. Some of these needs are related to areas that may be easily solved, such as facilitating preparation of specific types of food or facilitating the possibility to listen to songs, music, literature or TV programs representing a familiar and homely context. Allowing for this type of individual care might be crucial to families considering using formal support such as home-based services or nursing homes [16]. On the other hand, there might be more challenging differences, as exemplified by the health personnel and relatives who described different views in regard to life-prolonging treatment in advanced stages of dementia: views representing basic philosophical or moral differences in approaches to managing the last stage of life. Finally, challenges related to language; communication and understanding were put forward as the most common and fundamental barrier to provide adapted and dignified care within the framework of a nursing home, day care centre or home-based services.

### Loss of community, culture and language

Regardless of ethnicity, being placed in a residential care facility, such as a nursing home, implies different losses for all people: abstract losses such as loss of role, lifestyle, freedom, autonomy and privacy; material losses, such as loss of home and personal belongings; and social losses, including loss of contact with family, friends and pets [35]. As exemplified in this study, older people with immigrant backgrounds, as members of minority groups, will face additional challenges when adjusting to institutional living organized by the dominant culture. In addition to a potential loss of the care that was previously provided by the family, people with minority ethnic backgrounds may experience loss of culture and loss of community [31]. As exemplified in our study, food, drinks and care practises may be unfamiliar. In addition, there is a potential loss of music, literature, folklore, TV programs or religious rituals, and this loss can contribute to a great sense of isolation for the older person [16, 36, 37]. Another major cultural loss is that of not being able to communicate in the mother tongue. Both relatives and health personnel acknowledged how a lack of common language affects the ability to get to know the person with dementia, provide adjusted care and prevent social isolation. In a recent study on communication across linguistic and cultural diversity in a residential dementia care

setting in Sweden, observations showed that the residents who did not speak Swedish communicated less and tended to sit alone more than those who spoke Swedish [38]; similar to findings in our study. Also, earlier studies do show that minority ethnic groups in care facilities for older people tend to have less communication with the staff compared to people representing the majority group [39, 40]. This pattern is related not only to language but to differences in lifestyle and care practises between the health personnel working in the institutions and the residents representing minority ethnic groups [35, 36]. In another Swedish study [41], which included nursing staff, relatives and residents in a care setting for older Finns speaking Finnish, it was found that the common Finnish language provided the residents with a feeling of belonging to a group with shared identity. Sharing the language made the residents able to talk about how the war had affected their lives, and these experiences were known by health personnel and recognized by other residents by nodding or through comments. Also in the dementia ward, there were examples of knowledge of each other's past and present lives, such as residents reciting songs or nursery rhymes together; friendships developed based on what was interpreted as some sort of mutual understanding of community [41]. A scoping review [26] exploring the influence of language and culture in the caregiving of people with minority ethnic background living with dementia, underlines this strong relation between language and culture. In light of the tendency for people with different linguistic and cultural background to loose familiarity with the second language and cultural context of the country they are living in, the review concludes that "culture-linguistic" congruity is highly beneficial in securing the wellbeing of people with dementia; and that the lack of such congruity has the opposite effect. The authors underline that the cultural dimension is important to address since specific aspects of culture, such as religion, traditional customs and food, are important needs that can persist into the dementia process. However, through referring to the concept of linguistic relativity, culture is underlined as a similarly crucial dimension in regard to language; a relation mirrored in the concept of culture-linguistic congruity. Thus, as shown in this study, language and "culture" are not separate entities but are highly intertwined, and a lack of a common language can therefore make it difficult to communicate as well as establish trustful relations. In other words, older people from immigrant and minority ethnic groups may be expected to become familiar and adjust to a partly unfamiliar culture and language at a time when they are particularly vulnerable with regard to maintaining and confirming their sense of identity [42].

### Language and access to adjusted care

In line with the above arguments, our findings indicate that there is a risk that a person who is unable to speak the majority language might be misinterpreted, and thus changes in the clinical picture may remain undiscovered. Nurses described how changes in the ability to communicate; normally alerting them to consider clinical changes, could be overlooked in cases where language did not provide such information; suggesting therapeutic nihilism in the meeting with patients not speaking the majority language [43].

Other studies show that misunderstandings, misinterpretations, and inability to communicate in their own language may cause older persons to become more passive and dependent, with the consequence of deteriorating physical and mental health [36, 44]. In studies by Ekman et al. [45, 46] and Runci et al. [47–50], it was found that immigrants with a dementia disorder functioned far below their level of latent competence if the interaction was based on the person with dementia's second language rather than on their native tongue. Studies also show that such patients may be perceived to be more heavily affected by dementia than they are, which can increase the speed of the progression of the disorder due to the person not receiving linguistic inputs that are cognitively challenging [13]. Another issue related to

language is thematised in a recent study [44] exploring the relationship between agitation and people with dementia who speak English as their second language. The study showed that speaking English as the second language rather than as the first language was associated with significantly higher agitation level (measured by the Cohen-Mansfield Agitation Inventory). These findings are in accordance with the findings of our study; health personnel describing how being able to respond to agitated patients in their own language would calm down difficult situations. This was an experience so persistent that multilingual nurses or nurse assistants off duty were called even in the evening or at night to help calm down agitated patients.

According to Norwegian health legislation, Guidance on use of interpreter in health services and Guidelines on dementia care, an interpreter should be used when necessary in order to facilitate communication and secure that important information is shared and understood [51, 52]. However, as found in our study, the use of interpreters in long term care such as in nursing home and home-based care is not common due to financial and practical reasons.

## Past and present discontinuities

In general, several earlier studies conclude that for people with minority ethnic backgrounds, admission to care facilities creates many discontinuities in relation to customary lifestyle, daily life patterns, social networks and support. This situation relates not only to the change in physical localization and the process of institutionalization but also to the services being adjusted to the culture of the majority population [35, 39, 53, 54]. Leininger [55] has repeatedly highlighted this point of concern: practicing nursing solely based on the majority culture may cause cultural imposition; that is, practicing nursing in this way may impose cultural (ethnocentric) expressions of care, which would be harmful instead of therapeutic.

Nursing homes have played a pivotal role in Norwegian eldercare for decades and remain today a crucial institution in the care for frail older people. In general, eldercare is widely considered a public domain, and the relative importance of nursing homes in Norway can be illustrated by the fact that 43.3 percent of all deaths occur in nursing home institutions. Thus, native Norwegians are highly familiar with the use of public care institutions [56], in contrast to people with immigrant background still representing a rather young population, and that still not tend to use long term care facilities [2, 13]. In our study health personnel reported that a lack of common language or cultural frame of reference could manifest in difficulties communicating and interacting as well as a feeling (expressed by relatives) that one did not "fit in" or belong in the particular context. Lack of exposure to Norwegian culture and language may be caused by the older person coming to Norway in old age, been taken care of by their relatives for a long time before the decision to use public help was made, or having generally lived a life in close adherence to their own ethnic group's cultural and religious practises. It may also be explained by the fact that people with different cultural and linguistic backgrounds not only tend to mix languages, have difficulties in distinguishing between languages, or revert to speaking only their native language as the disorder progress, but also may forget the context in which they are living in at the moment [53]. In other words, people with minority ethnic backgrounds living with dementia may believe that they still live in the context in which they were brought up, a socio-cultural environment that may differ substantially from the present context [53]; a situation that may add to their confusion.

An interesting finding in our study was that some of the health personnel with immigrant background themselves perceived to be approached more often by the patients, even if they did not share language or culture; which possibly can be explained by a feeling of affiliation and trust due to a shared minority status. Being in such a minority position can be seen in the light of intersectionality theory [57], postulating that gender, ethnicity and class are social

positions expressed simultaneously in producing e.g. inequality in health and health care. It is suggested that this perspective is particularly important in immigrant health research where the acculturation paradigm dominates and where the exploration of how immigrant health trajectories are shaped by gender, social class and ethnicity is often absent [58]. Even though immigrant groups are diverse they often share certain features based on their status as immigrants; lower socio-economic status may create barriers to good health and health care, because they speak different languages they may experience language and health literacy barriers when seeking health care, and they may face discrimination and prejudice upholding their structural position in society [59]. Lack of familiarity with elements of the Norwegian culture and language is likely to be interlinked with ethnicity; level of education; job opportunities and experience; as well as gender roles; thus, creating differences in the possibility of accessing adjusted care. For example, in Norway, only 52 percent among persons with a refugee background are employed compared to 79 percent of the majority population; and immigrant women being less likely to work than men [60].

As illustrated in our study, living in an institution organized according to the frames of the culture, values and norms of the majority population may also cause difficulties in negotiating across religious- or culturally specific themes and values. This may potentially cause discontinuities regarding ethics; preferences and decisions related to the different aspects of treatment and care practises. Additionally, juridical and ethical conflicts concerning important decisions such as end-of-life decision-making may occur and create uncertainty in regard to which value system is to govern; conflicts that have been thematised in several other studies [61, 62]. In other words, receiving care in a context dominated by the majority culture may be challenging for patients and relatives, as it may create several discontinuities not only in daily life patterns and social networks but also in relation to religion and common values.

## Culturally congruent caring services

To address such discontinuities one solution could be to develop culturally congruent caring services, described by Leininger as care that are congruent, meaningful, and relevant to individuals and groups representing different cultures [63]. In a research briefing conducted by Moriarty et al. [37], different studies show that minority ethnic groups have different opinions on whether they prefer culturally specific or culturally mixed dementia services. However, even among those preferring "mixed" services, it is important that care is culturally acceptable in the form of food and care practices and that the atmosphere is one where people can feel "at home". For example, a study on dementia in the Bangladeshi diaspora in England, showed that a shortage of culturally and religiously appropriate services, made the family carers more likely to provide the care themselves [64]. In another study, also based on the Bangladeshi diaspora in England, it was shown that for many of the participants it was vital that the service providers and home carers had the same cultural and religious identity [65]. In Sweden, so-called ethnically profiled dementia care has emerged, and the policy draws on the acknowledgement that person-centred care should integrate people's different cultural and linguistic background; implying having the opportunity to practice one's religion, having access to culturally adjusted food, being able to maintain cultural customs and traditions, and being able to communicate with someone who speaks the same mother tongue as the person with dementia [53]. The rationale for ethnically profiled nursing homes derives from the acknowledgement that both language and cultural differences might give rise to specific care needs. Some studies from Sweden suggest that ethno-culturally profiled residential care facilitates the use of nursing homes among people with immigrant backgrounds and dementia. It is suggested that this is because relatives feel less guilty about not fulfilling their duty of filial piety when they can

ensure that the older person with dementia is taken care of by someone who speaks the same language and has the same cultural background [22, 41, 66]. However, as argued by Antelius et al. [22], research on dementia among minority ethnic groups is scarce, and the starting point has been an assumption that the "cultural ingredients" are in accordance with the needs and preferences of the different ethnic groups (but have never been tested with the target group). Thus, it is difficult to say to what degree and among what type of group members the different needs have actually been met [22]. This argument was developed further in a Scottish study, where most respondents (health personnel working with patients and relatives from South Asia) concluded that improvement of mainstream service would make the dementia services more accessible and that developing "dozens and dozens of different models to meet the needs of different subgroups" would be challenging [67].

Ethnically profiled nursing homes can be *one way* of maintaining person-centred care [53], as it could address the discontinuities experienced by people with another linguistic and cultural background. However, as noticed by Torres [68], the risk is that one may (again) be caught in the "otherness" trap by assuming that people sharing certain characteristics have similar needs and preferences and that those preferences represent "challenges" for those delivering the services. Not only may needs, preferences and opportunities differ between people with different religious backgrounds, ages, genders, social classes and the like (compare intersectionality), but ethno-cultural groups cannot be placed within a vacuum that does not change in response to the surrounding society [53, 69]. To what degree and in what direction such changes may occur is difficult to predict, as some patients or relatives may, to a large extent, have acculturated to the host society, while other patients or relatives may see the preservation of an ethno-cultural continuity as particularly important [53]. In a synthesis of qualitative studies on awareness and understanding of dementia among people from South Asia, the author underlines that the South Asian culture and associated religions are complex and varied. Thus, the numerous types of food, religions, languages, education systems, castes, employment statuses, gender rights and social statuses makes it impossible to make any generalizations regarding needs [69]. Similarly, Jutlla [23], based on her research review from the UK on ethnicity and cultural diversity in dementia care, warns about making assumptions based on generalized and stereotyped views from existing research on black and minority ethnic (BME) communities. There are differences both between and within groups and individuals, thus health care workers should practice a person-centred approach in the dementia care of BME people, recognizing the diversity between and within groups [23]. "Cultural competence", underlined as important by the health personnel in our study, is, according to Jutlla [23], a comprehension of such diversity and is characterized by a value-based perspective that acknowledges individuality. Similarly, as argued by Leininger, in order to provide culturally congruent care health personnel must uncover people's worldview, religion, family practices and norms, as well as the impact of environmental, political, social and economic factors in people's life [63]. In other words, factors that may have impact on an individual, social as well as a structural level.

This can be seen in the light of critical voices noticing that "cultural contexts" do not necessarily have to be ethnic but may be based on other types of communities, such as people coming from rural areas, having a particular level of education, belonging to a particular type of social class, or adhering to a specific religion [70]. It can be argued that in pursuing the development of care facilities aiming at adapting to specific ethnic groups may challenge the traditional definition of equality by a definition based on cultural particularity. In line with the theory of intersectionality [57, 58], such a focus may eventually pave the way for specific needs among patient groups with high socioeconomic status while risking that the emphasis on choice of lifestyle may conceal or justify emerging inequalities in care based on social class

[70]. Yet another critical input, in line with the views of the respondents in our study, is that existing surveys from Sweden, for example, have not shown any widespread wish to live in ethnically homogenous facilities [70, 71].

Subsequently, even though some studies from Sweden (e.g., 41) do show positive results, that does not necessarily mean that "culture-specific" nursing homes or day care centres are the only or best way of securing quality of life among people with an immigrant or minority ethnic background living with dementia. Independent of how one interprets and values the categories of ethnicity and culture, understanding people's life histories, experiences of migration, values related to culture and communities, and the possible impacts of those factors seems crucial in aiding patients and relatives to live well with dementia [23]. A biographical approach to needs assessment and care planning, directed at the distinct and individual life stories of the residents and their relatives, could facilitate dementia care where individual *as well as* socio-cultural needs are addressed [35]. Such an approach has been suggested as a way to bridge cultural differences, given that ethnicity is potentially a category fallacy, which may hinder tailored, person-centred care [72]. However, to be able to access individual as well as socio-cultural needs, time and familiarity seem to be indispensable resources. In a study on communication across linguistically and culturally diverse people with dementia in a residential care setting, the authors particularly underlined the importance of continuity in interpersonal relations [38]. Similar to the findings of our study, accumulation of knowledge about a person's needs, preferences, behaviour, verbal and non-verbal expressions, and what triggers or prevents agitation, is necessary to establish a common ground, and this common ground is proven to be particularly intricate in relations where one of the participants has dementia [73, 74]. In an exploratory study from Sweden, which included family caregivers and health personnel providing care for people with dementia who had lost their Swedish second language skills, it was found that the family members played a crucial role in establishing such common ground. Through facilitating communication, relatives made it possible for the persons living with dementia to recall memories and mediate specific needs; facilitating access to the cultural activities that they wanted and that the professional caregivers where not able to identify or deliver [27]. Also, health personnel in our study underlined the importance of an even closer collaboration with relatives if the patients with dementia had another linguistic or cultural background. A close relationship with relatives may also prevent misunderstanding and insecurity and facilitate solutions in situations where different value systems confront each other and must adjust. However, relatives are not always present, and many of the patients may not have relatives they can rely on. Thus, to secure that health personnel can provide adjusted and equitable care to a more diverse patient group, institutional constraints also need to be taken into account [75]; such as not being allocated more time or resources (e.g. use of interpreters) when getting to know and assess patients with another cultural and linguistic background.

## Strengths and limitations

The initial FGDs with older immigrants from 10 different countries were analyzed in the beginning of the study and served as an inspiration to identify themes and questions in consecutive interviews. We believe this initial familiarity with different ways of articulating and perceiving the phenomenon of dementia and dementia related care served as an important contribution to secure internal validity in the overall study. By triangulating data sources (different healthcare providers, relatives of people with dementia, older immigrants), healthcare settings (e.g., GP centres, nursing homes, day-care centres, home-based services, geriatric and psychiatric polyclinics, and hospital-based memory clinics), locations (different parts of Oslo, and six different counties in the northern, western, and southeast parts of Norway),

methods (FGDs and IDIs), and analysts (two researchers reading and analysing all the transcripts), we explored variations and contradictions as well as the consistency of different data sources. The number of participants is rather high to be a qualitative study, and a possible limitation is that we have not managed to provide thorough descriptions of all the different perspectives provided. Even though we had some contact with persons living with dementia through the relatives, we did not include any participants as we were unable to find any person that were able to communicate with us, mainly due to loss of language. This must be considered a weakness as this group of people are the once that have the direct experience of receiving care.

## Conclusions and implications

This study shows that there are needs related to language and culture that should be addressed in order to meet the health and care needs of older immigrants with dementia in Norway. These findings are in line with other studies, but there is lack of research on good practices in adapting dementia services to diversity. To meet the challenge of an ageing population and lack of health care personnel, and also increasing employment and integration, the Norwegian authorities have recruited and trained many immigrants to work in nursing homes and home-care services [8]. It seems crucial that the Norwegian health authorities systematically address how this linguistically and culturally diverse workforce can be used as an important resource, potentially reducing problems related to communication as well as the feeling of not belonging for patients with diverse backgrounds. How to best use a diverse workforce could be piloted inspired by existing knowledge and experiences, and play a role in intervention development and evaluation. Several participants among health personnel emphasized the need for "cultural competence" training of staff and in medical and nursing schools. However, focusing on building "cultural competence" risk imposing a generalised understanding of culture on patients [14]. Approaches based on building competencies in relation to ethnicity or culture of origin may overemphasize the cultural dimension of diversity, obscuring the role of gender, education, social class or caste [14]. Thus, to secure adjusted and equitable services health personnel might benefit from considering who is disadvantaged and in need of help to compensate for barriers related to language, ethnicity, gender or socio-economic position. On a structural level, it seems necessary to allocate more time and resources, including the use of interpreters when assessing and getting to know persons with another linguistic and cultural background; preferably formalised during e.g. admission to nursing homes and at regular intervals with the family and/or the patient, if needed.

To meet individual as well as socio-cultural needs and in line with cultural congruent care, a biographical approach, aimed at identifying distinct and individual life stories of the residents and their relatives, may be more fruitful than establishing care institutions or practises aimed at specific ethnic groups. With regard to achieving a biographical approach, most previous research on people with dementia who are bilingual or multilingual underlines the need for residents and staff to have a shared (native) spoken language. Health personnel also need to become familiar with each person's way of being ill, on both a cultural and an individual level, often expressed through the body language as well through the gradual changes occurring when living with progressive dementia. In other words, continuity in relation to language and culture is important, but continuity of care such as in relations, may be equally important to ensure that people with dementia with an immigrant background receive adjusted and equitable care. Future research should look more into both structures and conditions for- and practical ways of delivering cultural congruent dementia care.

## Acknowledgments

The authors thank the participants for sharing their experiences.

## Author Contributions

**Conceptualization:** Mette Sagbakken, Reidun Ingebretsen, Ragnhild Storstein Spilker.

**Formal analysis:** Mette Sagbakken, Ragnhild Storstein Spilker.

**Funding acquisition:** Ragnhild Storstein Spilker.

**Investigation:** Mette Sagbakken, Reidun Ingebretsen, Ragnhild Storstein Spilker.

**Methodology:** Mette Sagbakken, Reidun Ingebretsen, Ragnhild Storstein Spilker.

**Project administration:** Mette Sagbakken, Ragnhild Storstein Spilker.

**Supervision:** Mette Sagbakken.

**Validation:** Mette Sagbakken, Reidun Ingebretsen, Ragnhild Storstein Spilker.

**Writing – original draft:** Mette Sagbakken.

**Writing – review & editing:** Reidun Ingebretsen, Ragnhild Storstein Spilker.

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
