## [Decision Letter · Decision Letter 0]

28 Apr 2020

PONE-D-20-07935

How to adapt caring services to migration-driven diversity? A qualitative study exploring challenges and possible adjustments in the care of people living with dementia

PLOS ONE

Dear Professor Sagbakken,

Thank you for submitting your manuscript to PLOS ONE. As you will see below, three reviewers have considered it in detail and they have some reservations. Therefore, whilst we feel that it has merit, it does not fully meet PLOS ONE’s publication criteria as it currently stands.  We invite you to submit a revised version of the manuscript that considers the points raised during the review process.

We would appreciate receiving your revised manuscript by Jun 12 2020 11:59PM. To enhance the reproducibility of your results, we recommend that if applicable you deposit your laboratory protocols in protocols.io, where a protocol can be assigned its own identifier (DOI) such that it can be cited independently in the future. For instructions see: http://journals.plos.org/plosone/s/submission-guidelines#loc-laboratory-protocols

We look forward to receiving your revised manuscript.

Kind regards,

Antony Bayer

Academic Editor

PLOS ONE

Journal Requirements:

3. Please amend your manuscript to include your abstract after the title page.

Reviewers' comments:

Reviewer's Responses to Questions

**Comments to the Author**

1. Is the manuscript technically sound, and do the data support the conclusions?

Reviewer #1: Partly

Reviewer #2: Partly

Reviewer #3: Yes

2. Has the statistical analysis been performed appropriately and rigorously? 

Reviewer #1: N/A

Reviewer #2: N/A

Reviewer #3: N/A

3. Have the authors made all data underlying the findings in their manuscript fully available?

Reviewer #1: No

Reviewer #2: No

Reviewer #3: No

4. Is the manuscript presented in an intelligible fashion and written in standard English?

Reviewer #1: No

Reviewer #2: Yes

Reviewer #3: Yes

5. Review Comments to the Author

Reviewer #1: Review of manuscript: How to adapt caring services to migration-driven diversity? A qualitative study exploring challenges and possible adjustments in the care of people living with dementia

Thank you for the opportunity to review this paper that deals with and important and timely subject in many European countries. The paper explores challenges and possible adjustments related to receiving and providing public care for people from minority ethnic groups with dementia living in Norway, using a qualitative approach. Further, the authors aim at contributing to a debate on culturally congruent care based on their findings as well as existing research. The main findings are very much in line with existing literature, for instance language and communication barriers are found to be the main challenge for adjusted care and culturally/religious views may affect decisions about palliative care. Based on the results, the authors make an argument that a biographical approach to dement care may be preferable to culturally congruent services and highlight a culturally diverse workforce in mainstream services as an important resource. The authors conclude that continuity in relations are at least as important as continuity in relation to culture and language.

Overall, I find the manuscript to be a very long read. I acknowledge there are no word limit in PLOS ONE, but with approximately 11,500 words and several passages that go into too much detail or discussion, the manuscript should be reduced by at least 2000 words – preferable more - thereby significantly tightening up its focus and descriptions. Also, the authors need to make a stronger point about what contributes or adds to the existing literature. Finally, the manuscript needs a thorough readthrough to check English grammar and remove occasional type-o’s. Based on this, I cannot recommend publication of the manuscript in its current form.

Specific comments:

Introduction

General comment: The introduction includes very comprehensive descriptions and discussion of immigration and prevalent dementia in migrants in Norway, barriers in access to dementia care, approaches to culture-sensitive care, and the importance of language barriers. Some of this seem more suited for the discussion section or should preferable be significantly condensed.

Page 2

In the first paragraph, please use either “percent” of “%” for consistence.

In the second paragraph, dementia is referred to as a disease. This is not correct and should be changed to disorder or a similar term.

In the second paragraph, 3 lines from the bottom, “doctors” should be “medical doctors”.

Page 3

In the last paragraph, it is stated that patients “lack ability to use services”. This may unintentionally come out a bit ethnocentric, and this whole section seem to miss appoint about no culture-sensitive services being available. Could it be, that available services do not correspond to the needs of the patients, i.e. they “lack ability to provide services”?

Page 5

Is the second aim really of the study an aim or is it rather a discussion of the implications of the findings? It seems that culturally congruent services fit well under the first aim (possible adjustments to services). Please consider taking out this aim and discussing it as an implication of the findings instead.

Methods and sample strategy

General comments: The descriptions and details of this section are very comprehensive. I strongly suggest tightening up and condensing these descriptions, and preferably collapsing some of the many subsections.

Page 7

The whole section called “Design” seems a bit unconventional. Not only does it not cover the design of this qualitative study, i.e. Grounded Theory, Phenomenological, etc. or other details relevant to the design of this particular study, including the adherence to standard guidelines for qualitative research (SRQR or COREQ), which is required in PLOS ONE. It also lists the results of previous papers originating from the research programme that the study is part of. I strongly suggest removing this section and describing the study design in other parts of the Methods section. Also, all details about the previous papers from the research programme should be deleted and simply referred to by their references.

Page 8

I can be hard to get an overview of the participants in the different FGDs, DYIs and IDIs. Would it be possible to present the participant characteristics in one or two tables?

Page 10

In the section “Research team”, it is sufficient to specify the academic background of the researchers. Their employments will be irrelevant to most readers.

In the last line, the word “translator” should be “interpreter”.

Page 12

In the first paragraph, there is again reference to methods used in other articles from the research programme that seem irrelevant to the present study. Please delete this.

Also, a new aim is suddenly introduced here that was not described under aims in the introduction: to inform policy makers. Either include this as an aim of the study in the introduction or delete it entirely.

Page 13

Here you present a table that has no number and is not referred to in the text. Please add a Table heading and number and refer to in in the text.

In the Validity section, the final paragraph can be understood in the way that a high number of participants is a limitation in qualitative research? Please consider rephrasing.

Results

General comments: This section is also very long and covers several themes included in just four subsections. As a reader it can be hard to follow the red line in the subsections and I would suggest tightening up this section and adding additional subsection to the existing subsection to denote the individual themes identified.

Discussion

General comments: I can’t help but feel a that short discussion of the general lack culture-sensitive services in Norway is lacking in the discussion. I understand the authors would like to make an argument that a biographical/person-centered approach to dement care may be preferable to culturally congruent services, but I do no think this is entirely supported by the findings. One approach does not necessarily rule out the other. It may very well be that some service users may prefer culturally profiled services while others will not (which you also find). So maybe both options should be available? Also, it may be important to consider the results in light of the recruited informants. Could it be that older immigrants and relatives to people with dementia who a open to participating in this research would also be more open to mainstream services? Please add a limitation section where you discuss the limitations of the study, including the generalizability of the results.

Also, there is a lot of text explaining the findings from other studies. Could this possibly be condensed and shortened down.

Page 25

In line 2, it is mentioned that the study found challenges that appear challenging to patients. But the views of patients were not included in this study. Please also add this as a limitation.

Conclusion and Implications

Although it intuitively makes sense, the final remark about the importance of continuity on relations and not only language and culture does not seem to be supported by the results of the present study. This conclusion mainly rests on the results of other studies.

Reviewer #2: This is an interesting article, however I was not entirely sure what the conclusions were and how services should be shaped to better care for people with dementia from minority ethnic communities. I was surprised to find the persistent ‘othering’ of ‘immigrants’ throughout the article. ‘Immigrants’ (I am not convinced that this is the correct term) were portrayed against an assumed homogenised group of ‘native’ Norwegians without much qualification and demographic analysis. Even though, the literature quoted on immigrant health experiences is extensive, there was limited engagement with the literature on transcultural care, cross-cultural psychiatry and medical anthropology which could frame some of the issues raised. Furthermore, contrary to the article’s title and aims, there was limited to contribution to the field of health service design and no well thought out proposals for new services are presented at the end. Also, the recommendations on possible adjustments were assessed for feasibility and scalability. Finally, there appears to be some overlap with materials that have already been published elsewhere (BMC Health Services Research). The team is very knowledgeable and this is a very interesting and much-needed study but I would recommend further work on this paper.

Please see some suggestions below:

Introduction

1.The word ‘immigrants’ is used throughout without any qualification. I am not convinced this is the right term to use, certainly without offering justification.

2. Even though, there is a mention of diversity within the ‘immigrant’ category, the discourse used is ‘othering’ this group against an assumingly homogenous ‘native’ population. Is there any evidence that Norwegians for example are not expected to care for their elderly parents?

3. The literature review in the Introduction is extensive but is using very broad generalisations throughout. I would recommend specifying the ethnicity of the patients you are referring to and their country of residence when mentioning specific examples.

4. Similarly, the concept of ‘filial piety’ is mentioned as an exotic ‘other’. You explain a bit more about in your BMC paper but in this paper it is left uncontextualised. Are there no similar phrases in the Norwegian language? Was this concept explicitly mentioned by the participants in this study? What terms did people use?

Design

Good summary of previously published work.

Methods and sample strategy

It would be helpful to add some more information on how you recruited participants, our inclusion criteria and what strategies did you use to eliminate bias.

Analysis

You could add more info on analysis. Why did you choose the Kvale and Brinkman’s descriptions of levels of interpretation? What did you learn by using it?

Results

Receiving care in unfamiliar settings and surroundings

How many participants mentioned this? Can you provide more details about their background or circumstances to contextualise the quotes?

What type of ‘integration’ is needed in a care home? What does this mean? Add more details. Do you mean taking part in activities?

When the ‘immigrants’ were younger and worked, they operated in diverse workplaces, schools etc. They are presented like they are aliens who did not contribute to Norway’s economy and society. Is this the intention?

Also, was the relatives’ involvement only problematic? They provided free care and free-up professionals time.

Providing and receiving care in unfamiliar languages

Budget restraints are mentioned but google translate is free. Please add more context regarding what type of translations are allowed. Also, we know that non-verbal communication is very important in dementia care. Did noone mention anything about using non-verbal communication to overcome linguistic barriers?

Views on culturally profiled caring services

Xenophobia among other care home residents hinted here but not elaborated. Did you have any more evidence that care homes are hostile environments towards people from minority ethnic backgrounds?

You make some suggestions are offered on making the care homes more welcoming places but the data is very limited. Do you have more data to expand this paragraph and the one below on competence needs? My understanding from the title and abstract was these two sections would be key.

Competence needs

Is the same as the para above? Or does it need to be a separate section?

Discussion

The emphasis appears to be on individuals, not on the sociocultural context in which these individuals are embedded in. Assumptions like ‘lack of exposure to Norwegian culture’ are not contextualised. I’d like to see more analysis on whether this apparent lack of exposure is a result of dementia and loss of recent memories and if not, some analysis on the context that produced this ‘lack of exposure’.

There was no discussion on how participant ethnic background, occupation or gender shaped opinions.

Conclusions and Implications

I’d like to see more focus on how to apply findings from this study on service design.

Reviewer #3: Comments to the authors:

Thank you for the opportunity to review this article. The topic of dementia among the ethnic minority community in Norway is an important one where of course more research is needed. This is an interesting paper; however, the article cannot be published in this current form. More works need to be done in order to strengthen the paper. Such as:

Authors need to update the references: I have noticed references were used from the years of 1983, 1986, 1991, these references are very old, not helpful to be honest. I have also noticed some papers were cited from the authors Jutlla, MacKenzie, Uppal etc. But it would be good if you could update most recent publications in your references. Please try to include citations to the papers of authors in your immediate field that have been published very recently, year 2020. A list might be useful:

• Hossain, M.Z., Stores, R., Hakak, Y., Dewey, A. (2020). Traditional gender roles and effects of the dementia caregiving within a South Asian ethnic group in England. Dementia and Geriatric Cognitive Disorders. doi: 10.1159/000506363

• Hossain, M.Z., Stores, R., Hakak, D., Crossland, J., & Dewey, A. (2020). Dementia knowledge and attitudes of the general public among the Bangladeshi community in England: a focus group study. Dementia and Geriatric Cognitive Disorders. doi: 10.1159/000506123

• Hossain, M.Z., Khan, H.T.A. (2020). Barriers to access and ways to improve dementia services for a minority ethnic group in England. J Eval Clin Pract. 2020;1–9. https://doi.org/10.1111/jep.13361

• Hossain, M.Z., & Khan, H. T. A. (2019). Dementia in the Bangladeshi diaspora in England: A qualitative study of the myths and stigmas about dementia. Journal of evaluation in clinical practice.

• Hossain, M.Z., Crossland, J., Stores, R., Dewey, A., & Hakak, Y. (2018). Awareness and understanding of dementia in South Asians: A synthesis of qualitative evidence. Dementia.

# The authors mentioned about immigrant backgrounds of living with dementia in Norway many times. I was disappointed to see more details about the immigrants' background. As a reader, I would like to see what their backgrounds were, what are their religions (most important thing) etc. Probably, a demographic table would be more interesting to see.

# Abstract: an extra space added before the full stop of the first sentence

# First sentence of page number 3 is vague, not clear. Please re-write it, breaking down it would make more sense, I think.

# On page 3, However, many older immigrants come from societies where family structures and expectations of care from family members are stronger than in many Western societies (10, 11).

Sentences like this are not helpful, too general, can you give any example?

# Page 3, Ethnic differences in the use of dementia care services have been documented in Norway and internationally. Needed a reference for this statement

# Page 5, The authors underline that the cultural dimension is important to address since specific aspects of culture, such as religion, traditional customs and food, are important needs that can persist into the dementia process.

I'm not quite sure about this sentence, however, religion may or may not be the part of a culture, it's debatable, perhaps a clarification was needed?

# Page 7, I agree with the authors on this statement: Women in particular, might feel obliged to conform to a traditional caregiver role, but without the support from a wider extended family, and in the context of other pressing roles and duties.

I feel a reference is needed with this above sentence and there is a recent paper on the ethnic minority's traditional gender role caring issues: please see below, might be useful:

• Hossain, M.Z., Stores, R., Hakak, Y., Dewey, A. (2020). Traditional gender roles and effects of the dementia caregiving within a South Asian ethnic group in England. Dementia and Geriatric Cognitive Disorders. doi: 10.1159/000506363

# Page 10, The research team: I am not sure about how putting all the research team's bio-data, education details and academic positions would be helpful to the readers. I rather feel too much personal information have been included there.

# Page 11, a semi-structure guide was used: How did you develop a semi-structure guide, based on what information or studies did you prepare your interview questions?

# Previous studies elsewhere found that for Muslim carers or people with dementia religion play a big part.... what about their religion, presumably, some of these participants were Muslims and Islam put a great concern about care giving and receiving? Such as

# Details needed about who interviewed the female Muslim participants, there was an issue of interviewing Muslim women found in the literature, you may need to consult with the article where the male author could not interview the female carers: Hossain, M. Z., & Khan, H. T. A. (2019). Dementia in the Bangladeshi diaspora in England: A qualitative study of the myths and stigmas about dementia. Journal of evaluation in clinical practice.

# Discussion needs to be tightened up:

# Perhaps, need to create a section for Recommendation for research & policy: Research is beneficial for any community, what would be the recommendation for the future research and policy makers; perhaps how can this research be useful for ethnic minority community in Norway?

# Limitation & Conclusion: a separate limitation and conclusion would be useful for the readers.

6. PLOS authors have the option to publish the peer review history of their article (what does this mean?). If published, this will include your full peer review and any attached files.

Reviewer #1: Yes: T. Rune Nielsen

Reviewer #2: No

Reviewer #3: No

---

## [Author Response · Author response to Decision Letter 0]

9 Sep 2020

Dear Editor, 

Below are the responses listed according to the different reviewers input. In the manuscript the changes are marked in either red or trough track changes: In areas were the text/content are radically rewritten (including text being removed or added) the text is marked in red. In areas where we only have removed content/passages this can be seen through track changes. 

We do hope that the revisions made will make the manuscript suitable for publication in PLOS ONE.

Best regards Mette Sagbakken, corresponding author 

 

Reviewer #1: Review of manuscript: How to adapt caring services to migration-driven diversity? A qualitative study exploring challenges and possible adjustments in the care of people living with dementia

Thank you for the opportunity to review this paper that deals with and important and timely subject in many European countries. The paper explores challenges and possible adjustments related to receiving and providing public care for people from minority ethnic groups with dementia living in Norway, using a qualitative approach. Further, the authors aim at contributing to a debate on culturally congruent care based on their findings as well as existing research. The main findings are very much in line with existing literature, for instance language and communication barriers are found to be the main challenge for adjusted care and culturally/religious views may affect decisions about palliative care. Based on the results, the authors make an argument that a biographical approach to dement care may be preferable to culturally congruent services and highlight a culturally diverse workforce in mainstream services as an important resource. The authors conclude that continuity in relations are at least as important as continuity in relation to culture and language.

Overall, I find the manuscript to be a very long read. I acknowledge there are no word limit in PLOS ONE, but with approximately 11,500 words and several passages that go into too much detail or discussion, the manuscript should be reduced by at least 2000 words – preferable more - thereby significantly tightening up its focus and descriptions. Also, the authors need to make a stronger point about what contributes or adds to the existing literature. Finally, the manuscript needs a thorough read through to check English grammar and remove occasional type-o’s. Based on this, I cannot recommend publication of the manuscript in its current form.

General comments: The manuscript is long, and thus one of the reasons we chose PLOS ONE, having no word limit. We have condensed certain parts of the introduction, parts of the results as well as the discussion. However, the two other reviewers have asked us to add content several places, as well as more empirical and theoretical references, and the length of the manuscript is therefore approximately the same after the revision. 

Regarding contribution to existing literature: throughout the discussion we have tried to make it clearer how our findings contribute and support existing literature. One of the main reasons, however, for doing this research, is that it was commissioned and financed by the Norwegian Directorate of Health, as research on this group of people/patients living in Norway was almost nonexistent at the time. The Directorate wanted research to be carried out for better health service planning adjusting to the needs of a growing group of older immigrants in Norway; thus the main rationale and the main contribution was to explore the experiences in the Norwegian context in particular. 

Our manuscript is language edited by Wiley Editing Services (see attachment) and we have gone through the manuscript again; editing some minor errors.

Specific comments:

Introduction

General comment: The introduction includes very comprehensive descriptions and discussion of immigration and prevalent dementia in migrants in Norway, barriers in access to dementia care, approaches to culture-sensitive care, and the importance of language barriers. Some of this seem more suited for the discussion section or should preferable be significantly condensed.

Author: We have shortened parts of the introduction and moved some of the content to the discussion part.

Page 2

In the first paragraph, please use either “percent” of “%” for consistence.

In the second paragraph, dementia is referred to as a disease. This is not correct and should be changed to disorder or a similar term.

Author: this had been corrected.

In the second paragraph, 3 lines from the bottom, “doctors” should be “medical doctors”.

Author: this had been corrected.

Page 3

In the last paragraph, it is stated that patients “lack ability to use services”. This may unintentionally come out a bit ethnocentric, and this whole section seem to miss appoint about no culture-sensitive services being available. Could it be, that available services do not correspond to the needs of the patients, i.e. they “lack ability to provide services”?

Author: In page 3, we have already addressed this dimension, by the sentence: “However, current care services are undoubtable designed for a more homogenous group of people, and it is argued that this poses challenges to equitable services” (14). 

We have however removed this part of the sentence (ability to use services) and added yet another sentence where we emphasize that one of the causes of services not being used is lack of ability to adjust services to different patient groups. 

Page 5

Is the second aim really of the study an aim or is it rather a discussion of the implications of the findings? It seems that culturally congruent services fit well under the first aim (possible adjustments to services). Please consider taking out this aim and discussing it as an implication of the findings instead.

Author: We agree that this second aim could be seen as an implication of the findings and have thus removed it from this paragraph. 

Methods and sample strategy

General comments: The descriptions and details of this section are very comprehensive. I strongly suggest tightening up and condensing these descriptions, and preferably collapsing some of the many subsections.

Author: We have shortened down/rewritten the methods chapter by addressing the below comments. However, one of the reviewers wanted us to add more information about the overall design as well as the sample strategy, thus some information is also added. 

Page 7

The whole section called “Design” seems a bit unconventional. Not only does it not cover the design of this qualitative study, i.e. Grounded Theory, Phenomenological, etc. or other details relevant to the design of this particular study, including the adherence to standard guidelines for qualitative research (SRQR or COREQ), which is required in PLOS ONE. It also lists the results of previous papers originating from the research programme that the study is part of. I strongly suggest removing this section and describing the study design in other parts of the Methods section. Also, all details about the previous papers from the research programme should be deleted and simply referred to by their references.

Author: We have shortened and made changes to the description of the design/methods and included a description of methodology underpinning the research. 

It is quite common and often requested to briefly present results from other papers that are part of larger studies; to allow transparency and to help the reader to get an overall view of the results and how they complement each other. The presentation of the results was in this review appreciated by the second reviewer. To be able to respond to the different views of reviewer one and two we have therefore chosen to shorten the description of the result, by only referring to the topic addressed in the separate papers. 

Page 8

I can be hard to get an overview of the participants in the different FGDs, DYIs and IDIs. Would it be possible to present the participant characteristics in one or two tables?

Author: We have structured (by the use of headlines) and condensed the information about the participants to make it more accessible. 

Page 10

In the section “Research team”, it is sufficient to specify the academic background of the researchers. Their employments will be irrelevant to most readers.

In the last line, the word “translator” should be “interpreter”.

Author: We have limited the information about the researchers in this section. In the last line the word translator is replaced. 

Page 12

In the first paragraph, there is again reference to methods used in other articles from the research programme that seem irrelevant to the present study. Please delete this.

Also, a new aim is suddenly introduced here that was not described under aims in the introduction: to inform policy makers. Either include this as an aim of the study in the introduction or delete it entirely.

Author: We have deleted the part where we refer to methodological/theoretical consideration done in the other papers. We have also deleted what may be interpreted as a new aim (inform policy makers). We have left the description of the initial FGDs with older immigrants, as these data served as an important inspiration for identifying themes and questions in consecutive interviews (IDIs and FGDs), and helped contextualize data in the following interviews; thus being a part of the analytical process. 

Page 13

Here you present a table that has no number and is not referred to in the text. Please add a Table heading and number and refer to in in the text.

In the Validity section, the final paragraph can be understood in the way that a high number of participants is a limitation in qualitative research? Please consider rephrasing.

Author: We have added a table heading and number. Regarding the description of validity, this is a well-known concern, as the analysis in most qualitative research is concerned about analyzing and presenting data in depth, including contextual information/details. We have tried to rephrase this to make this point clearer. However, this part is now moved to the end of the discussion, called “Strength and limitations”.

Results

General comments: This section is also very long and covers several themes included in just four subsections. As a reader it can be hard to follow the red line in the subsections and I would suggest tightening up this section and adding additional subsection to the existing subsection to denote the individual themes identified.

Author: It is a bit difficult to relate to comments pointed to the manuscript being too long/passages/section being too long without the reviewer having any suggestions on where to cut. We have tried to tight it up but have also added some data (the need for continuity) due to the concern of one of the other reviewers. We have added some additional subsections to split some of the themes. 

Discussion

General comments: I can’t help but feel a that short discussion of the general lack culture-sensitive services in Norway is lacking in the discussion. I understand the authors would like to make an argument that a biographical/person-centered approach to dement care may be preferable to culturally congruent services, but I do not think this is entirely supported by the findings. One approach does not necessarily rule out the other. It may very well be that some service users may prefer culturally profiled services while others will not (which you also find). So maybe both options should be available? Also, it may be important to consider the results in light of the recruited informants. Could it be that older immigrants and relatives to people with dementia who a open to participating in this research would also be more open to mainstream services? Please add a limitation section where you discuss the limitations of the study, including the generalizability of the results.

Also, there is a lot of text explaining the findings from other studies. Could this possibly be condensed and shortened down.

Author: We have condensed some of the findings from other studies in the discussion. We have added a Strength and limitation section at the end of the paper, where we address different strengths and limitations. Since we added groups of immigrants (9 focus groups) that were recruited to talk about health services for older immigrants – irrespective of having relatives with dementia/using existing services (in addition to the other participants with more direct experience) – we believe we have captured the essence of opinions of a large and varied part of the immigrant population. In the findings section there are also examples of participants that favored services having a cultural profile; we are referring to that of a majority of participants not having such preferences, not everyone. Therefore, we do not see the need to address this as a limitation. 

We have tried to make the discussion more nuanced in regard to the positive aspects of facilitating culturally congruent services. 

Page 25

In line 2, it is mentioned that the study found challenges that appear challenging to patients. But the views of patients were not included in this study. Please also add this as a limitation.

Author: We have added this in the limitation part. 

Conclusion and Implications

Although it intuitively makes sense, the final remark about the importance of continuity on relations and not only language and culture does not seem to be supported by the results of the present study. This conclusion mainly rests on the results of other studies.

Thank you for this important input. The need of continuity does not rest on the results of other studies. All participants frequently thematised this; both indirectly and directly. However, we see now that this needs to be emphasized more in the results and have included more data/focus on this in the results. The conclusion/suggestions is thus directly based on the empirical findings, supported by other studies.

Reviewer #2: This is an interesting article, however I was not entirely sure what the conclusions were and how services should be shaped to better care for people with dementia from minority ethnic communities. I was surprised to find the persistent ‘othering’ of ‘immigrants’ throughout the article. ‘Immigrants’ (I am not convinced that this is the correct term) were portrayed against an assumed homogenised group of ‘native’ Norwegians without much qualification and demographic analysis. Even though, the literature quoted on immigrant health experiences is extensive, there was limited engagement with the literature on transcultural care, cross-cultural psychiatry and medical anthropology which could frame some of the issues raised. Furthermore, contrary to the article’s title and aims, there was limited to contribution to the field of health service design and no well thought out proposals for new services are presented at the end. Also, the recommendations on possible adjustments were assessed for feasibility and scalability. Finally, there appears to be some overlap with materials that have already been published elsewhere (BMC Health Services Research). The team is very knowledgeable and this is a very interesting and much-needed study but I would recommend further work on this paper.

Author: General comments on the literature: 

The articles we refer to in the manuscript are the result of a thorough literature review conducted (ordered by the Norwegian Directorate of Health), and even though we might have included even more studies (with different types of perspectives) we believe that the references are quite extensive. Since one of the reviewers have suggested 5 concrete articles he wants us to add, and the other reviewer wants us to cut and condense the paper, we have chosen not to not add further references. 

We do however refer to transcultural care through Leninger’s well-known work, and we have – based on your input – chosen to elaborate more on her particular work in the discussion part. 

Regarding overlap: we have removed one part of the results (about relatives involving/moving in to nursing homes to participate in intimate care of their relatives) based on your input on this section being overlapping. 

The rest of the comments addressed above, will be answered below:

Please see some suggestions below:

Introduction

1.The word ‘immigrants’ is used throughout without any qualification. I am not convinced this is the right term to use, certainly without offering justification.

Author: In this paper, we use “minority ethnic groups” when we refer to studies from outside Scandinavia, as this is the most commonly used term in these papers. However, in the parts of the manuscript where we refer to Norwegian statistics, Norwegian/ Scandinavian studies, and the present project, we use the term “Immigrant”, which is the terminology used by Statistics Norway, and is the most commonly used term in academic and public discourse. Importantly, the use of “Immigrant” also indicates that we are not referring to our native population, such as the Saami’s. This information is now added in page 6.

2. Even though, there is a mention of diversity within the ‘immigrant’ category, the discourse used is ‘othering’ this group against an assumingly homogenous ‘native’ population. Is there any evidence that Norwegians for example are not expected to care for their elderly parents?

Author: As emphasized in page 3, we do underline that “Immigrants’ experiences of aging and health are diverse as they belong to different socio-economic, ethnic, cultural and linguistic groups and have varying levels of education and work experience. Further, they have various reasons for migrating, and have lived in Norway for different lengths of time (9). Family structures and relationships are diverse both within and between different immigrant groups. However, many older immigrants come from societies where family structures and expectations of care from family members are stronger than in Western societies (10, 11).” We have also emphasized this point more in the discussion. 

To be able to understand the institutional practice in Norway in a better way we have also added relevant information about this in the discussion part (condensed version of the following text), page 28-29. 

In general, nursing homes have played a pivotal role in Norwegian eldercare for decades, and remain today an exceptionally important institution in regards to care for the elderly. In general, elderly care is widely considered a public domain, and the relative importance of e.g. nursing homes in Norway can be illustrated by the fact that 43.3 percent of all deaths occur in nursing home institutions. Thus, native Norwegians are highly familiar with the use of public care institutions, in contrast to people with immigrant background still representing a rather young population. The particular topography and geography of Norway should also be taken into account when considering the importance of nursing homes. Norway is relatively sparsely populated over a large (relative to number of inhabitants) geographical area, has few large cities (most of which serve as regional centers), and has a topography and infrastructure making traveling long distances a challenge in many parts of the country. Many Norwegians therefore live far from hospitals and other parts of the specialized health service, both in distance and in travel time, giving the nursing homes a vital local function, as well as being a significant local employer. In 2009 the total number of care recipients in long-term beds in Norwegian nursing homes was 34 800. This figure amounts to almost 1 percent of the total population, far more than any comparable country. Many elderly Norwegians live in nursing homes and the majority of these elderly also die in the nursing homes. Among residents occupying what is labelled a ≪long-term bed≫, it is estimated that 95 percent die while residing in the institutions (Ågotnes, G. (2017). The Institutional Practice: On nursing homes and hospitalizations. Cappelen Damm Akademisk/NOASP (Nordic Open Access Scholarly Publishing).

3. The literature review in the Introduction is extensive but is using very broad generalisations throughout. I would recommend specifying the ethnicity of the patients you are referring to and their country of residence when mentioning specific examples.

Author: the reason why there seem to be rather broad generalizations in the introduction is that we rely mainly on review articles that summarize patterns in barriers experienced by people with another linguistic or cultural background. In the introduction reference nr 15-17, 19-20, 23-26 are all narrative or systematics reviews. When referring to primary studies we are specifying which ethnic groups that are in focus.

4. Similarly, the concept of ‘filial piety’ is mentioned as an exotic ‘other’. You explain a bit more about in your BMC paper but in this paper it is left uncontextualised. Are there no similar phrases in the Norwegian language? Was this concept explicitly mentioned by the participants in this study? What terms did people use?

Author: the concept of filial piety is used in the introduction as a frame (also based on review articles) to explain different types of caring patterns. There are no similar phrases in the Norwegian language and since filial piety is a theoretical concept, the respondents did not use this word. However, the concept is suited to explain feelings of commitment, reciprocity and other descriptions used by the participants when describing how and why they felt that they needed to provide care for their family members. We have added information about the common and accepted use of institutional care for older people in Norway in the discussion part, which is rather exceptional even when compared to Scandinavian countries. 

Design

Good summary of previously published work.

Methods and sample strategy

It would be helpful to add some more information on how you recruited participants, our inclusion criteria and what strategies did you use to eliminate bias.

Author: During the process of recruitment, we found it very difficult to find relatives as well as health personnel who had direct experience with caring for patients living with dementia representing another linguistic and cultural background (see reference 2, 13). Thus, as now explained clearer in the text, we had to rely on a convenience sampling, were snowballing within already existing network and within local care services was one of the strategies. We have elaborated more on each of the sampling processes under “Sampling strategy.”

Regarding bias: We used extensive time and resources to find enough – and different participants to participate in this study; at a later stage also involving many different geographical parts of this long-stretched country. Bias was reduced by triangulating data sources (healthcare providers, relatives of people with dementia, older immigrants); locations (different parts of Oslo, and six different counties in the northern, western, and southeast parts of Norway); healthcare settings (e.g., GP centres, nursing homes, day-care centres, home-based services, geriatric and psychiatric polyclinics, and hospital-based memory clinics); methods (FGDs and IDIs); and analysts (two researchers reading and analyzing all the transcripts). Within and across all these data sources, we explored variations and contradictions as well as the consistency of different data sources. 

Analysis

You could add more info on analysis. Why did you choose the Kvale and Brinkman’s descriptions of levels of interpretation? What did you learn by using it?

Author: One of the reviewers suggested condensing this part. However, we have added a short explanation of the main function of moving between different interpretational contexts under “Analysis”: These three interpretational contexts derive from different explications of the researcher’s perspective and lead to different levels of analysis, and serve to make explicit the analytic questions posed to a statement.

Results

Receiving care in unfamiliar settings and surroundings

How many participants mentioned this? Can you provide more details about their background or circumstances to contextualise the quotes?

Receiving care in unfamiliar settings is a headline that covers several aspects that was, in different ways, mentioned by a large group of respondents. We have tried to contextualize this part of the results in a better way and have rewritten large parts of this particular section. 

We have chosen not to provide more details about the participant’s background as it is few people with immigrant background living with dementia in Norway, and relatives as well as health personnel working in this particular field may easily be recognized. As an example, in one of our previous articles one of the reviewers paid attention to the fact that he thought he knew the identity of one of the participants among the doctors; this based on his well-known opinions/experiences expressed in a quote in the paper. 

What type of ‘integration’ is needed in a care home? What does this mean? Add more details. Do you mean taking part in activities?

Author: Thank you for this input. We have rewritten this part and angled it without too much focus on integration; rather that of being in unfamiliar surroundings due to exposure to the majority culture only. 

When the ‘immigrants’ were younger and worked, they operated in diverse workplaces, schools etc. They are presented like they are aliens who did not contribute to Norway’s economy and society. Is this the intention?

Author: No, this is not the intention, and we have added more context to bring forward the content in a better way. See among other page 30.

Also, was the relatives’ involvement only problematic? They provided free care and free-up professionals time.

Author: No, this was of course not only problematic. The section where we present results related to relatives engaging in the (intimate) care is removed, as it partly overlaps with the article published in Qualitative Health Research. The section where we present findings related to differences in the view of care practices (related to later stages/terminal care) is instead elaborated, as it illustrates how different religious views and/or care practices may conflict ethically as well as legally. 

Providing and receiving care in unfamiliar languages

Budget restraints are mentioned but google translate is free. Please add more context regarding what type of translations are allowed. 

Author: We have already mentioned that health personnel used google translate as well as personal with corresponding language competencies as strategies in their daily work in the result chapter. This is marked in the text. Since we mainly talk about long-term care, using qualified interpreters was not/is not common in the daily care due to financial and practical reasons. We have added some more information about this in the discussion part, page 28.

Also, we know that non-verbal communication is very important in dementia care. Did noone mention anything about using non-verbal communication to overcome linguistic barriers?

Author: thank you for this important input. Yes, there was several examples of health personnel talking about body language. This was related to the importance of continuity in care; where the emphasize was on how continuity was important to get to know the patients, including their body language, over time. We have added some of these quotes in the results chapter. 

Views on culturally profiled caring services

Xenophobia among other care home residents hinted here but not elaborated. Did you have any more evidence that care homes are hostile environments towards people from minority ethnic backgrounds?

Author: It was only one participant that mentioned this directly. Even if she was the only one, we included this quote as it might represent the experience of others. We have also addressed this indirectly in the first paragraphs (now rewritten) where we point to how the surroundings may feel alienating due to an exposure (songs, food, music, TV programs) to the majority culture only. 

You make some suggestions are offered on making the care homes more welcoming places but the data is very limited. Do you have more data to expand this paragraph and the one below on competence needs? My understanding from the title and abstract was these two sections would be key.

Author: We have added more in the paragraph now called “lack of continuity of care”, which refers to the importance of getting to know people and their individual preferences and clinical changes to facilitate a more homely and safe environment. We also make the need for familiar environment a stronger point in the beginning of the discussion part. 

Competence needs

Is the same as the para above? Or does it need to be a separate section?

Author: we have chosen to bring the last two sections together, as reviewer one wants us to condense the discussion, and as pointed to by you – these topics can be presented together. 

Discussion

The emphasis appears to be on individuals, not on the sociocultural context in which these individuals are embedded in. Assumptions like ‘lack of exposure to Norwegian culture’ are not contextualised. I’d like to see more analysis on whether this apparent lack of exposure is a result of dementia and loss of recent memories and if not, some analysis on the context that produced this ‘lack of exposure’.

Author: We have elaborated more on this in the discussion part, and we have also rewritten part of this to make the content clearer. See page 29.

There was no discussion on how participant ethnic background, occupation or gender shaped opinions.

Author: We did not see any clear pattern related to a particular ethnic background; and occupation and religion was not asked for but came up when relevant (religion in particular) during the interviews/FGD. This study did not seek to examine and compare different groups’ experiences. Rather – through variation in the sample; e.g. participants from more than 18 countries, representing different “positions” (older immigrants, relatives to persons with dementia, different health personnel) in regards to the topic being explored – we have sought to identify variations and similarities; including patterns of consistency trough different data sources. However, we do discuss how ethnic background, occupation or gender may shape peoples experiences and opinions by the use of the theoretical perspective of intersectionality; and trough that we emphasize that the interplay or “sum” of many different positions is likely to be more important than the different factors such as gender, age, education and ethnicity alone. 

Conclusions and Implications

I’d like to see more focus on how to apply findings from this study on service design.

Author: Our study has not focused on service design per se; it has focused on how it is possible to adjust public care in relation to people living with dementia with another linguistic or cultural background. In this part of the study, we have focused on both day care centers, homebased services, and nursing homes (as well as GP centres, geriatric and psychiatric polyclinics, hospital-based memory clinics in the articles already published) – all with quite different services and focus. Thus, we have no ambition to suggest different health services design regarding all these different treatment and care institutions, but rather provide an insight into different experiences and views represented by different/relevant actors on how services can be adapted in a better way. 

Our conclusion is also drawn on the basis that we did not see any clear patterns in needs that would justify a specific service design, but rather there is need to become familiar with each individual and their specific and often changing needs due to progressive dementia. Further, due to people in this group having another linguistic background (and often lose their second language first) it is important to invest in linguistic and cultural diversity among the health workers. We have rewritten the conclusion to emphasize the main point in a better way.

Reviewer #3: Comments to the authors:

Thank you for the opportunity to review this article. The topic of dementia among the ethnic minority community in Norway is an important one where of course more research is needed. This is an interesting paper; however, the article cannot be published in this current form. More works need to be done in order to strengthen the paper. Such as:

Authors need to update the references: I have noticed references were used from the years of 1983, 1986, 1991, these references are very old, not helpful to be honest. I have also noticed some papers were cited from the authors Jutlla, MacKenzie, Uppal etc. But it would be good if you could update most recent publications in your references. Please try to include citations to the papers of authors in your immediate field that have been published very recently, year 2020. A list might be useful:

• Hossain, M.Z., Stores, R., Hakak, Y., Dewey, A. (2020). Traditional gender roles and effects of the dementia caregiving within a South Asian ethnic group in England. Dementia and Geriatric Cognitive Disorders. doi: 10.1159/000506363

• Hossain, M.Z., Stores, R., Hakak, D., Crossland, J., & Dewey, A. (2020). Dementia knowledge and attitudes of the general public among the Bangladeshi community in England: a focus group study. Dementia and Geriatric Cognitive Disorders. doi: 10.1159/000506123

• Hossain, M.Z., Khan, H.T.A. (2020). Barriers to access and ways to improve dementia services for a minority ethnic group in England. J Eval Clin Pract. 2020;1–9. https://doi.org/10.1111/jep.13361

• Hossain, M.Z., & Khan, H. T. A. (2019). Dementia in the Bangladeshi diaspora in England: A qualitative study of the myths and stigmas about dementia. Journal of evaluation in clinical practice.

• Hossain, M.Z., Crossland, J., Stores, R., Dewey, A., & Hakak, Y. (2018). Awareness and understanding of dementia in South Asians: A synthesis of qualitative evidence. Dementia.

Author: As part of this rather large research project, we also conducted a literature review. Some of the sources are old, but still represent relevant perspectives, and where therefore included; e.g. some of the still relevant perspectives of Leininger M. (Becoming aware of types of health practitioners and cultural imposition. Journal of Transcultural Nursing, 1991; 2(2), 32-39.)

As the literature review was done in 2015-2017 and the main work with the article was done in 2019, we have not been familiar with the above mentioned studies, but have now included those being of relevance. However, some of the studies that are suggested are focused on the perceptions and aetiology of dementia, and would have suited better in our already published study on this: Sagbakken, M, Spilker, RS, Ingebretsen, R. (2019). Understanding dementia in ethnically diverse groups: a qualitative study from Norway. Ageing & Society 

# The authors mentioned about immigrant backgrounds of living with dementia in Norway many times. I was disappointed to see more details about the immigrants' background. As a reader, I would like to see what their backgrounds were, what are their religions (most important thing) etc. Probably, a demographic table would be more interesting to see.

Author: We have yet no records of the number (prevalence) and country background of immigrants living with dementia in Norway. Regarding the participants in this study, they are under the methods section described with country of origin, age and gender. The position of interest is people with a different linguistic and cultural background in need of care due to cognitive impairment (not diagnosed) and dementia; and the experiences of those providing relevant services (relatives and health personnel). We did not ask about people’s religious background, but religion came up as part of their cultural background whenever relevant in responding to the questions. We have restructured the section under methods where we present the participants making the information about them more accessible. 

# Abstract: an extra space added before the full stop of the first sentence.

Author: this has been done 

# First sentence of page number 3 is vague, not clear. Please re-write it, breaking down it would make more sense, I think.

Author: The sentence is rewritten. 

# On page 3, However, many older immigrants come from societies where family structures and expectations of care from family members are stronger than in Western societies (10, 11).

Sentences like this are not helpful, too general, can you give any example?

Author: We have added an example to give more meaning to the content. 

# Page 3, Ethnic differences in the use of dementia care services have been documented in Norway and internationally. Needed a reference for this statement

Author’: References are inserted.

# Page 5, The authors underline that the cultural dimension is important to address since specific aspects of culture, such as religion, traditional customs and food, are important needs that can persist into the dementia process.

I'm not quite sure about this sentence, however, religion may or may not be the part of a culture, it's debatable, perhaps a clarification was needed?

Author: We have added religion as a separate element in this paragraph. 

# Page 7, I agree with the authors on this statement: Women in particular, might feel obliged to conform to a traditional caregiver role, but without the support from a wider extended family, and in the context of other pressing roles and duties.

I feel a reference is needed with this above sentence and there is a recent paper on the ethnic minority's traditional gender role caring issues: please see below, might be useful:

• Hossain, M.Z., Stores, R., Hakak, Y., Dewey, A. (2020). Traditional gender roles and effects of the dementia caregiving within a South Asian ethnic group in England. Dementia and Geriatric Cognitive Disorders. doi: 10.1159/000506363

Author: The reference to this quote was our own already published paper. However, we have removed this sentence as one of the other reviewers wanted us to shorten down the description of earlier findings. 

# Page 10, The research team: I am not sure about how putting all the research team's bio-data, education details and academic positions would be helpful to the readers. I rather feel too much personal information have been included there.

Author: We have limited the information about the researchers in this section.

# Page 11, a semi-structure guide was used: How did you develop a semi-structure guide, based on what information or studies did you prepare your interview questions?

Author: The questions in the interview guide used in the initial FGD’s with healthy older adults were inspired by a thorough literature review that was part of the overall study commissioned by the Norwegian Directorate of Health. The questions in the interview guides used for relatives and health personnel were partly inspired by the same literature review, partly by perspectives provided by the first FGDs with older adults, and partly by discussions of experiences with experts in the field (health personnel, researchers, nongovernmental organization representatives) who served as a resource group throughout the research period. This information is now added.

# Previous studies elsewhere found that for Muslim carers or people with dementia religion play a big part.... what about their religion, presumably, some of these participants were Muslims and Islam put a great concern about care giving and receiving? Such as

Author: The reviewer has not completed the sentence/question, so it is difficult to respond properly. We did not ask about religion in particular but considered it as a part of the cultural background. The only time we could see a pattern of religious influence, was when life-preserving treatment was discussed; people with a Muslim background seemingly representing the group that had most difficulties in accepting treatment to be terminated. 

# Details needed about who interviewed the female Muslim participants, there was an issue of interviewing Muslim women found in the literature, you may need to consult with the article where the male author could not interview the female carers: Hossain, M. Z., & Khan, H. T. A. (2019). Dementia in the Bangladeshi diaspora in England: A qualitative study of the myths and stigmas about dementia. Journal of evaluation in clinical practice.

Author: The three authors, who are all females, did the interviews/FGD’s. We did not experience any problems in relation to interview either men or women. We have added information about the authors themselves performing the interviews under the paragraph Research team. 

# Discussion needs to be tightened up:

Author: We have tried to tighten the discussion, but also added references as suggested. 

# Perhaps, need to create a section for Recommendation for research & policy: Research is beneficial for any community, what would be the recommendation for the future research and policy makers; perhaps how can this research be useful for ethnic minority community in Norway?

Author: we believe there is a need for intervention research as we now have gained a rather broad knowledge base in regard to this group of patients. This is mentioned in the Conclusion.

# Limitation & Conclusion: a separate limitation and conclusion would be useful for the readers.

Author: We have ended the discussion with a paragraph addressing Strengths and Limitations and keeping a separate Conclusion/implication part.

---

## [Decision Letter · Decision Letter 1]

26 Nov 2020

How to adapt caring services to migration-driven diversity? A qualitative study exploring challenges and possible adjustments in the care of people living with dementia

PONE-D-20-07935R1

Dear Dr. Sagbakken,

I am sorry there has been some delay in getting back to you but we’re pleased to inform you that your manuscript has now been judged scientifically suitable for publication and will be formally accepted for publication once it meets all outstanding technical requirements.

Kind regards,

Antony Bayer

Academic Editor

PLOS ONE

Additional Editor Comments (optional):

Reviewers' comments:

Reviewer's Responses to Questions

**Comments to the Author**

1. If the authors have adequately addressed your comments raised in a previous round of review and you feel that this manuscript is now acceptable for publication, you may indicate that here to bypass the “Comments to the Author” section, enter your conflict of interest statement in the “Confidential to Editor” section, and submit your "Accept" recommendation.

Reviewer #1: All comments have been addressed

Reviewer #2: All comments have been addressed

2. Is the manuscript technically sound, and do the data support the conclusions?

Reviewer #1: (No Response)

Reviewer #2: Yes

3. Has the statistical analysis been performed appropriately and rigorously? 

Reviewer #1: (No Response)

Reviewer #2: N/A

4. Have the authors made all data underlying the findings in their manuscript fully available?

Reviewer #1: (No Response)

Reviewer #2: No

5. Is the manuscript presented in an intelligible fashion and written in standard English?

Reviewer #1: (No Response)

Reviewer #2: Yes

6. Review Comments to the Author

Reviewer #1: (No Response)

Reviewer #2: The authors have adequately addressed the reviewers comments on an individual basis but altogether the paper hasn't significantly improved. It is very long and very descriptive, perhaps the data would be best presented as two or three different papers with a clearer focus. The approach is sound and the findings important but the analysis needs more work.

7. PLOS authors have the option to publish the peer review history of their article (what does this mean?). If published, this will include your full peer review and any attached files.

Reviewer #1: **Yes: **T. Rune Nielsen

Reviewer #2: No

---

## [Editor Report · Acceptance letter]

11 Dec 2020

PONE-D-20-07935R1 

How to adapt caring services to migration-driven diversity? A qualitative study exploring challenges and possible adjustments in the care of people living with dementia  

Dear Dr. Sagbakken:

I'm pleased to inform you that your manuscript has been deemed suitable for publication in PLOS ONE. Congratulations! Your manuscript is now with our production department. 

Kind regards, 

on behalf of

Professor Antony Bayer 

Academic Editor

PLOS ONE